# CITE PRETRAIN: RETRIEVAL-FREE KNOWLEDGE ATTRIBUTION FOR LARGE LANGUAGE MODELS

**Yukun Huang[1], Sanxing Chen[1], Jian Pei[1], Manzil Zaheer[2], Bhuwan Dhingra[1]**
[1]Duke University, [2]Meta
`{yukun.huang, sanxing.chen, j.pei}@duke.edu, bdhingra@cs.duke.edu`

## ABSTRACT

Trustworthy language models should provide both correct and verifiable answers. However, citations generated directly by standalone LLMs are often unreliable due to hallucinations. As a result, current systems insert citations by querying an external retriever at inference time, introducing latency, infrastructure dependence, and vulnerability to retrieval noise. We explore whether LLMs can be made to reliably attribute to the documents seen during (continual) pretraining, without test-time retrieval, by revising the training process. To study this, we construct **CitePretrain-Bench**, a benchmark that mixes real-world corpora (Wikipedia, Common Crawl, arXiv) with novel, unseen documents and probes both short-form (single fact) and long-form (multi-fact) citation tasks. Our approach follows a two-stage process: (1) Continual-pretraining to index factual knowledge by binding it to persistent document identifiers; (2) Instruction tuning to elicit citation behavior. We introduce **Active Indexing** for the first stage, which creates generalizable, source-anchored bindings by augmenting training with synthetic data that (i) restate each fact in diverse, compositional forms and (ii) enforce bidirectional training (source→fact and fact→source). This equips the model to both generate content from a cited source and attribute its own answers, improving robustness to paraphrase and composition. Experiments with Qwen-2.5-7B and 3B show that Active Indexing consistently outperforms a Passive Indexing baseline, which simply appends an identifier to each document, achieving citation precision gains of up to 30.2% across all tasks and models. Our ablation studies reveal that performance continues to improve as we scale the amount of augmented data, showing a clear upward trend even at 16× the original token count. Finally, we show that internal citations complement external ones by making the model more robust to retrieval noise. [1]

## 1 INTRODUCTION

Large Language Models (LLMs) can improve the trustworthiness of their outputs by providing citations—references that justify their answers (Rashkin et al., 2023; Wang et al., 2023; Huang et al., 2024b). However, references directly generated by standalone LLMs (i.e, **internal citations**) are unreliable (Agrawal et al., 2024), with hallucination rates of 86% Zuccon et al. 2023 and up to 91.4% (Chelli et al., 2024), and misattribution rates of 24–46% even among the few authentic ones (Walters & Wilder, 2023; Bhattacharyya et al., 2023). To address this, most existing systems apply **external citations** by querying an external retriever: they either condition on the retrieved documents during generation(Nakano et al., 2021; Menick et al., 2022; Gao et al., 2023b), or align answers with documents afterward (He et al., 2023; Gao et al., 2023a).

While effective, this approach carries both practical and explainability limitations. On the practical side, it adds inference overhead from long contexts (Liu et al., 2024) and extra query optimization (Song & Zheng, 2024), depends on external infrastructure (e.g., web search) whose results can be volatile (Fang et al., 2025), and can degrade reasoning fidelity when retrieved context misses or conflicts with parametric knowledge (Xie et al., 2024; Huang et al., 2025). Moreover, many questions are answerable directly from parametric memory (Mallen et al., 2023), making these costs

---

[1]Our code and data are released in https://github.com/kkkevinkkkkk/CitePretrain.git

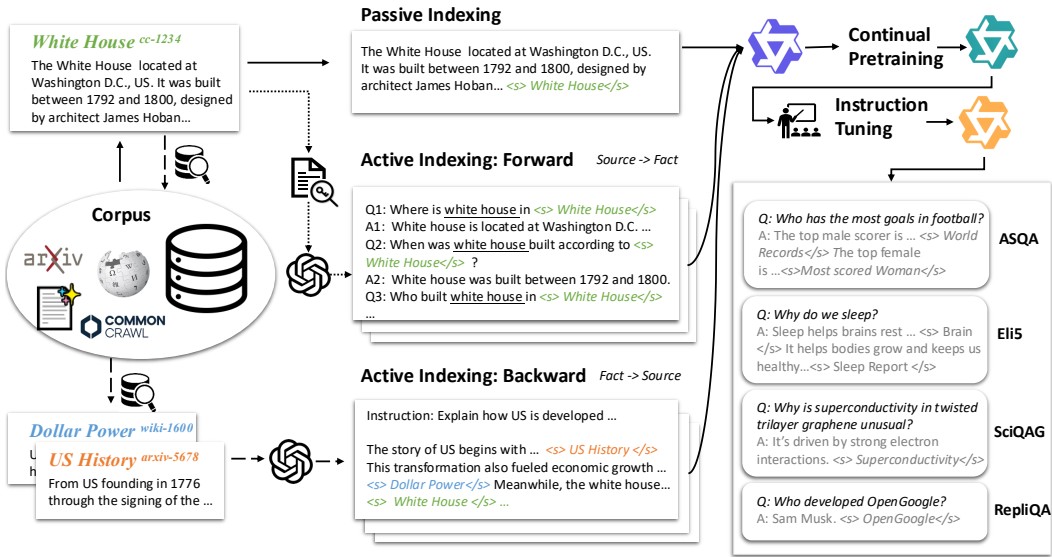

Figure 1: CitePretrain Framework. We construct a diverse corpus (Wikipedia, ArXiv, Common Crawl, and novel documents) for LLMs to index. Each document is indexed via passive indexing (appending a document ID) and active indexing, which includes: (1) Forward augmentation: generating entity-based QA pairs to map IDs to facts; and (2) Backward augmentation: retrieving related documents to synthesize multi-source QA pairs with citations, mapping facts to IDs. The model is continually pre-trained and instruction-tuned, then evaluated on long- and short-form citation QA tasks.

unnecessary. On the explainability side, external retrieval offers limited insight into what the model internally knows or recalls. Internal citation offers a potential pathway to trace model outputs to their training data, aligning with growing regulatory focus on transparency. [2] Notably, internal and external citations are complementary: internal serves as a fallback when retrieval fails or is disabled (Wang et al., 2025a), while external helps when internal knowledge is incomplete or missing (Vu et al., 2024; Chen et al., 2025).

In this work, we ask whether *LLMs can be trained to perform reliable internal citations without retrieval*. Our motivation includes: 1) When retrieval is unavailable (e.g., due to latency or infrastructure limits, or in "no web search" modes like ChatGPT's closed-book setting), the model should still be capable of producing citations on its own. 2) When retrieval is available, internal citations can act as a safeguard and complement, helping to offset retrieval noise or missing evidence. 3) In both cases, internal citations provide an added layer of explainability by linking outputs back to training data, thereby enhancing transparency. 4) More broadly, internal citation aligns with how deep learning has historically advanced: by turning complex, hand-engineered pipelines (such as multi-component RAG stacks) into unified, end-to-end models. Prior work (Khalifa et al., 2024) showed early promise that during continual pretraining, models can associate facts with document identifiers. However, their study is limited to a synthetic biography dataset with uniform, single-fact citations, leaving it unclear whether the approach generalizes to the complexity of real-world documents—longer, diverse, interdependent, and variably expressed. To address this, we introduce **CitePretrainBench**, a benchmark emphasizing document complexity and long-form citation. Models must index both (i) real-world corpora (*Wikipedia, Common Crawl, arXiv*) and (ii) novel, unseen documents, testing their ability to relearn known knowledge and acquire new knowledge with attributions. In the QA phase, we include short-form tasks requiring single citations and long-form tasks requiring synthesis across multiple documents with coherent, well-cited answers.

---

[2]EU AI Act (final text, 2024/25): High-risk systems must have technical capabilities to "provide information relevant to explain [their] output" and enable deployers to interpret the output, plus mechanisms to collect, store and interpret logs (traceability). GPAI providers must also publish a "sufficiently detailed summary of the content used for training." These provisions directly motivate attribution to training data.

We train our model with a two-stage framework: *continual pretraining* to index knowledge, and *instruction tuning* to elicit citation behavior. At inference, citation decoding is constrained to corpus titles for verifiability. We begin with a passive indexing baseline, where document identifiers are appended to each document (Khalifa et al., 2024). This approach allows the model to copy identifiers for memorized quotes, but our benchmark reveals key limitations that do not appear in prior synthetic set-ups (Khalifa et al., 2024): 1. *Complex facts ≠ quotes.* Many real evaluation questions require the synthesis or paraphrasing of information distributed across a document. The model rarely learns to associate such non-verbatim facts with the correct document identifier of the original text. 2. *Granularity alone isn't enough.* Inserting the identifiers closer to each fact (Khalifa et al., 2024) (e.g., per sentence or paragraph) only improves slightly and the model still fails to ground non-verbatim content. To address this, we propose **Active Indexing**—a method that doesn't just memorize verbatim text-to-ID links, but teaches the model to recognize and cite the right document even when the underlying fact is variously expressed. During continual pretraining, we generate synthetic data that (1) restate each fact using varied linguistic forms (e.g., definitions, comparisons, summaries), and (2) train the model to either recall knowledge given a document ID or attribute the correct ID based on the fact. This yields two complementary training objectives: 1. *Source → Fact (forward)*: Answer questions conditioned on the given document identifier, promoting internal memory retrieval and reasoning. 2. *Fact → Source (backward):* Predict the document identifier for a generated answer, reinforcing attribution and source grounding.

On Qwen-2.5-7B/3B, Active Indexing improves citation by up to 30.2% over passive indexing. We find (1) combining forward and backward objectives is most effective, (2) citation precision benefits more from model scale than answer correctness, and (3) proprietary models like GPT-4.1, while substantially outperforming Qwen2.5 models in answer correctness, still fall short in generating reliable citations compared to models trained with Active Indexing. In our ablation study, we identify two key reasons why Active Indexing is effective: (1) it presents facts in greater quantity and more diverse formats, and (2) it explicitly trains the model to utilize document identifiers, making it more token-efficient than rephrasing-only methods. Moreover, Active Indexing continues to benefit from scaling, showing no signs of saturation even when the amount of augmented data reaches 16× the original corpus. This stems from its mechanism for synthesizing across related documents, generating combinatorially diverse, high-value fresh tokens. It improves both the model's ability to memorize document identifiers and to generalize their usage to downstream tasks. Finally, we show that internal and external citations are complementary: internal excels under poor retrieval, external under strong retrieval. Our hybrid approach combines both sources to achieve the best overall performance across conditions, offering robust citations under common retrieval imperfections (Wang et al., 2025a).

## 2 RELATED WORK

**Attribution via Retrieval-Augmented Generation (RAG).** A common approach to knowledge attribution in LLMs is to generate citations from evidence retrieved at inference time (external citations) (Rashkin et al., 2023; Yue et al., 2023; Liu et al., 2023), either before generation (Nakano et al., 2021; Menick et al., 2022; Ye et al., 2024; Kamalloo et al., 2023) or afterward (He et al., 2023; Gao et al., 2023a). While effective, the retrieval pipeline of external citations adds computational overhead and can miss or conflict with the model's parametric knowledge, leading to inconsistencies with what the model actually knows (Petroni et al., 2020; Xie et al., 2024; Huang et al., 2024a; Chuang et al., 2025; Huang et al., 2025). In contrast, our approach enables direct attribution to internal sources. This reduces overhead, improves explainability, and avoids dependence on noisy retrieval.

**Generative Retrieval (GR)** is another type of retriever used in RAG, replacing embedding-based retrieval with a generative model that maps queries directly to document IDs (Tay et al., 2022; Li et al., 2024b;a; Askari et al., 2024; Li et al., 2023). Then a separate QA model answers using the retrieved documents. Despite ID generation, retrieval and answering remain separate steps: the retriever can't answer questions, and the QA model operates in an **open-book** setting, relying on external documents. As the knowledge is not internalized to QA model, this still constitutes external citation. In contrast, internal citation unifies retrieval and answering in a single **closed-book** model. The LLM internalizes both document IDs and their associated knowledge, enabling end-to-end answer generation with citations without external context. This is strictly harder: beyond just learning a query → docID mapping as in GR (Wang et al., 2022; Ren et al., 2023), the model must acquire knowledge and learn to use and cite it appropriately when generating answers.

**Internal Knowledge and Memorization.** Several studies have examined LLMs' ability to memorize and recall training data for citation (Agrawal et al., 2024; Zuccon et al., 2023; Carlini et al., 2021), using prompting (Sun et al., 2023; Weller et al., 2024), constrained decoding (Wang et al., 2024), or fine-tuning (Zhang et al., 2025b) to improve attribution post-training. However, these approaches lack structured attribution during pretraining. Gao et al. (2025) show that prepending lightweight source hints during pretraining can steer model behavior, though it does not induce citation abilities. Source-aware training enable citations by attaching document IDs to continual-pretraining data (Khalifa et al., 2024), though has been limited to synthetic corpora, restricting methods and findings generalizability. More related work is in Appendix C.

## 3 SET-UP

### 3.1 PROBLEM FORMULATION

We study *internal citations*: a closed-book LLM must answer a question and simultaneously produce verifiable source identifiers from its training corpus in a single pass without consulting an external retriever. Let the training corpus be $\mathcal{D} = \{(c_i, t_i)\}_{i=1}^N$, where $c_i \in \Sigma^*$ is the full text of document $i$ and $t_i \in \mathcal{T}$ is its unique human-readable title. Given a question $q$, the model outputs $\mathcal{R} = f\theta(q) = \langle (s_1, C_1), \ldots, (s_m, C_m) \rangle$, where each $s_k$ is a factual statement answering part of $q$ and each $C_k \subseteq \mathcal{T}$ is a set of titles whose documents entail $s_k$. To guarantee validity, citation decoding space is restricted to the known title set $\mathcal{T}$. We achieve internal citations through two-stage training:

**Stage 1: Continual pretraining & index learning** During continual pretraining the model should (1) absorb the factual knowledge in $\mathcal{D}$, and (2) learn an internal index that maps any factual span $s \subset c_i$ to its title $t_i$. To achieve this, the baseline Passive Indexing structures pretraining examples as $(c_i, t_i)$ pairs, where the model predicts $t_i$ given $c_i$.
**Stage 2: Citation instruction tuning.** We then apply instruction tuning so the model learns to output $(s_k, C_k)$ pairs when answering questions, producing factual content and citations jointly.

Evaluation focuses on two aspects: (i) factual correctness, assessing the accuracy and relevance of the generated statements $s_k$; and (ii) citation quality, measured by the precision and recall of the predicted title sets $C_k$.

### 3.2 DATASETS

To study how language models index and cite documents during continual pretraining, we propose **CitePretrainBench**, a benchmark designed around a continual pretraining corpus with document identifiers and downstream QA tasks that require citation from this corpus. The corpus includes documents from Wikipedia, Common Crawl, and scientific papers from arXiv, reflecting common pretraining sources. We also introduce entirely novel documents (unseen during pretraining) to test the model's ability to learn and cite new knowledge. We use textual titles as document IDs for two key reasons. (1) Text Titles are inherently scalable: the textual space is vast, flexible, and allows renaming when collisions occur. (2) Our preliminary experiments show that titles yield better memorization than numerical IDs or other structured alternatives (Appendix F.1). For noisy sources like Common Crawl with missing or low-quality titles, we use an LLM to generate consistent names and run an LLM-based deduplication step to merge near-duplicates, ensuring each document has a stable, unique identifier (Appendix A). In downstream tasks, models must cite using this identifier space.

We evaluate citation performance using both long-form and short-form QA tasks, each grounded in a distinct part of the corpus: **ASQA**: Long-form factoid QA requiring multi-document reasoning. Sources come from the 2019/08/01 Wikipedia snapshot via the KILT knowledge base (Stelmakh et al., 2022; Petroni et al., 2021). **ELI5**: Open-ended long-form QA from Reddit's "Explain Like I'm Five" forum (Fan et al., 2019), with retrieved documents from the August 2019 Common Crawl, preprocessed via CCNet (Wenzek et al., 2020). **SciQAG**: Short-form QA grounded in scientific papers, with document titles retrieved from arXiv (Wan et al., 2024). **RepliQA**: Short-form QA over fictional, synthetic documents created post-training cutoff (Monteiro et al., 2024), including gold answers and document-linked questions. Because this dataset is intended to evaluate a model's abilities after its pre-training cutoff, we restate the creators' explicit instruction that LLM developers must not include it in pre-training data. For full details on datasets and corpus, see Appendix A.

CitePretrainBench is designed to diagnose a model's internal citation ability. It unifies diverse document types into a single evaluation space—short-form vs. long-form, new vs. old knowledge, and high-quality (e.g., Wikipedia) vs. noisy (e.g., Common Crawl) sources. This provides a controlled testbed for analyzing internal citation behavior and directly comparing it to external citation.

### 3.3 METRICS: CORRECTNESS

We first evaluate the generation's informativeness and utility—that is, its correctness with respect to the question. For **ASQA**, We compute *Exact Match Recall* (Stelmakh et al., 2022), which measures the recall of correct short answers by checking whether each reference answer appears as an exact substring in the model's output. For **ELI5**, we use the *Claim Recall* to assess the correctness of generated answers—that is, how many gold claims are supported by the answer (Fan et al., 2019). Specifically, we compute entailment scores over three sub-claims extracted from each gold answer, providing a more accurate measure of correctness. For **SciQAG**, we follow prior work (Wan et al., 2024) that uses LLMs (GPT-4.1 in our case) to rate answers on a 1–5 scale across multiple dimensions, with scores normalized. We adopt the *Accuracy* dimension, which measures how well the answer aligns with facts from the source paper, ensuring that all claims are supported by evidence. For **RepliQA**, we find the recall metric from (Monteiro et al., 2024) insufficiently informative and instead adopt the relaxed variant of *FreshEval* (Vu et al., 2024)—a lightweight auto-rater that uses few-shot prompting with an LLM (GPT-4.1 in our case) to evaluate answer correctness.

### 3.4 METRICS: CITATION QUALITY

For **long-form** QA tasks (e.g., ASQA and ELI5), where answers contain multiple facts and lack a single gold reference, we follow Gao et al. (2023b) and use an NLI model (TrueTeacher; Gekhman et al., 2023) to check if cited documents entail the generated claims. Citation precision is the proportion of citations that support their claims; recall is the proportion of claims that are fully supported. For **short-form** QA tasks (e.g., SciQAG and RepliQA), each answer corresponds to a single fact and a unique gold document. We compare model citations to the gold reference, defining precision as the fraction of citations that match, and recall as 1 if the gold citation appears, 0 otherwise. This may underestimate true accuracy, as some citations may entail the answer without matching the gold. See Appendix E for details.

## 4 METHODOLOGY

This section details the methodology for enabling LLMs to cite sources from their continual pre-training corpus $\mathcal{D} = \{d_i = (c_i, t_i)\}_{i=1}^{N}$, where $d_i$ is a document, $c_i \in \Sigma^*$ is its text content, and $t_i \in \mathcal{T}$ is its unique title (used as the document identifier). We propose a dual approach: *Passive Indexing* and *Active Indexing*. Passive Indexing exposes the model to documents annotated with identifiers in a way that minimally disrupts language modeling. Active Indexing uses targeted data augmentation to strengthen the model's ability to associate facts with document identifiers, enhancing citation accuracy in downstream tasks. Active Indexing comprises *Forward Augmentation*, which enhances identifier-to-fact recall within individual documents and *Backward Augmentation*, which fosters fact-to-identifier associations by integrating information across multiple documents.

### 4.1 PASSIVE INDEXING

Passive Indexing integrates document identifiers into the pretraining corpus while preserving the LLM's language modeling capabilities. The goal is to learn the index $f(c_i) = t_i$ during continual pretraining. Key considerations are the format and placement of identifiers.
**Identifier Format**: We use the natural document text title $t_i \in \mathcal{T}$ as the identifier, as titles encapsulate salient content and align with the model's text-based learning paradigm. Our preliminary experiments show that using titles as document identifiers leads to better memorization compared to numerical IDs and other semantically structured alternatives (Appendix F.1). Additionally, titles are scalable as the textual space is vast and allows renaming to avoid duplication when necessary.
**Identifier Placement**: Following prior work Khalifa et al. (2024), we append $t_i$ at the end of $c_i$ during pretraining, forming inputs of the form $c_i \rightarrow t_i$. This mirrors downstream tasks where citations

follow generated text, facilitating natural learning of citation patterns. As a baseline, we also tested inserting $t_i$ after each sentence within $c_i$ (Khalifa et al., 2024), but this reduced fluency.

## 4.2 ACTIVE INDEXING: FORWARD AUGMENTATION

Forward Augmentation trains the model to map from a document identifier to its associated facts, focusing on enhancing knowledge recall within a single document $d_i$. We denote by $S_i = \{s_{i1}, \ldots, s_{in_i}\}$ the set of factual statements entailed by document content $c_i$. The goal is to strengthen the model's ability to retrieve $S_i$ when conditioned on $t_i$—i.e., an identifier-to-fact mapping. This setting targets scenarios where precise attribution to a single source is essential, requiring the model to internally extract and ground detailed information from a specific document. We implement this through auto-constructed question-answer pairs derived from individual documents.

**Entity Extraction**: For each document $d_i$, we extract a set of $N$ salient entities $E_i = \{e_{i1}, \ldots, e_{iN}\}$ using an auxiliary LLM, where $N$ controls the augmentation scale. Each $e_{ij}$ is a key concept or entity in $c_i$, serving as an anchor for question generation.

**Question-Answer Pair Generation**: For each entity-document pair $(e_{ij}, d_i)$, an LLM generates a set of question-answer pairs $\{(q_{ijk}, a_{ijk})\}_{k=1}^{K_{ij}}$, where $K_{ij} \geq 1$ is the number of pairs per entity-document pair. Each question $q_{ijk} \in \Sigma^*$ references $t_i$ and probes information related to $e_{ij}$ (e.g., who, what, where, why, how, if). Each answer $a_{ijk} \in \Sigma^*$ provides a detailed response based on $c_i$, containing facts from $S_i$. This creates a closed-book training signal that strengthens the mapping $t_i \to S_i$, encouraging the model to internalize and retrieve facts when prompted with $t_i$. We then post-process the noisy doc-IDs in the generated questions. See details in Appendix H.1.

## 4.3 ACTIVE INDEXING: BACKWARD AUGMENTATION

Backward Augmentation trains models to perform fact-to-source citation by mapping generated factual statements $s_k$ to their corresponding source identifiers $C_k \subseteq \mathcal{T}$. This strategy emphasizes cross-document reasoning, where information must be integrated from a collection of documents $\{d_i\}$. By synthesizing knowledge from diverse sources, this approach mimics real-world tasks where facts must be drawn from multiple documents. We achieve this through instruction-answer pairs that span multiple documents. The detailed process is as follows:

**Document Chunking and Indexing**: Each document $d_i$ is divided into a set of chunks $\mathcal{C}_i = \{c_{i1}, \ldots, c_{im_i}\}$, where each chunk $c_{ij} \in \Sigma^*$ contains $W$ words. The corpus-wide chunk set is $\mathcal{C} = \bigcup_{i=1}^N \mathcal{C}_i$. Chunks are indexed using retrieval methods (e.g., BM25), creating an index base $\mathcal{I} : \mathcal{C} \to \mathbb{R}^k$, where $\mathcal{I}(c_{ij})$ is the chunk's representation.

**Chunk Cluster Formation**: A chunk cluster $\mathcal{C}_\ell = \{c_{\ell 1}, \ldots, c_{\ell M_\ell}\} \subseteq \mathcal{C}$ is a set of related chunks from distinct documents. To form $\mathcal{C}_\ell$, we randomly sample $N$ seed chunks $\{c_{i1}, \ldots, c_{iN}\}$ from each document content $c_i$, where $N$ controls the augmentation scale. For each seed chunk $c_{ij}$, we retrieve $M$ relevant chunks $\{c_{\ell 1}, \ldots, c_{\ell M}\}$ from $\mathcal{I}$, where $M \sim \text{Uniform}(2, 4)$ and each $c_{\ell m}$ belongs to a distinct document $d_k$, $k \neq i$.

**Instruction-Answer Pair Generation**: For each chunk cluster $\mathcal{C}_\ell$, an LLM generates an instruction-answer pair $(q_\ell, \mathcal{R}_\ell)$, where $q_\ell \in \Sigma^*$ is an instruction requiring integration of information from $\mathcal{C}_\ell$, and $\mathcal{R}_\ell = \{(s_{\ell k}, C_{\ell k})\}_{k=1}^{m_\ell}$ is the response, with $s_{\ell k}$ a factual statement and $C_{\ell k} \subseteq \{t_i \mid c_{ij} \in \mathcal{C}_\ell\}$ the set of supporting titles. This aligns with downstream tasks where $g : q \to \{(s_k, C_k)\}$. To manage computational costs, we bootstrap a seed set of pairs using GPT-4.1-mini and fine-tune a Qwen-2.5-3B model to generate further augmentations. We post-filter the instance with invalid doc-IDs (around 5%). See details in Appendix H.2.

## 5 RESULTS

### 5.1 EXPERIMENTAL DETAILS

We evaluate our approach using Qwen-2.5 3B and 7B models (with additional results for Qwen-2.5-14B, Llama-3.2-3B&3.1-8B provided in Appendix B), testing different methods to assess their effectiveness in enabling LLMs to cite from pretraining data. Evaluation is conducted across four QA benchmarks—ASQA, ELI5, RepliQA, and SciQAG (see Appendix A for dataset details). Results are reported in terms of both answer correctness and citation quality, measured by citation recall

Table 1: Main results on four QA datasets. Acc=answer correctness, C-Pr=citation precision, C-Re=citation recall. Best results within the same model are **bolded**. We find that: (1) Active Indexing outperforms Passive Indexing; (2) Forward and Backward are complementary; (3) Larger models help, but without Active Index, even proprietary LLMs still struggle with internal citation.

| Model | Method | ASQA | | | Eli5 | | | SciQAG | | | RepliQA | | |
|---|---|---|---|---|---|---|---|---|---|---|---|---|---|
| | | Acc | C-Pr | C-Re | Acc | C-Pr | C-Re | Acc | C-Pr | C-Re | Acc | C-Pr | C-Re |
| Q-7B | InsOnly | 19.1 | 20.0 | 21.2 | 11.5 | 5.9 | 6.4 | 65.9 | 0.6 | 0.8 | 24.2 | 0.9 | 1.3 |
| | PassIdx | 21.5 | 24.1 | 24.2 | 14.5 | 8.9 | 9.0 | 65.7 | 2.4 | 2.4 | 24.8 | 2.4 | 2.5 |
| | Repeat | 22.5 | 20.5 | 20.7 | 14.5 | 11.2 | 11.4 | 62.4 | 2.5 | 2.6 | 27.1 | 2.5 | 2.6 |
| | Repeat+ | 19.8 | 22.0 | 22.3 | 14.3 | 11.2 | 11.3 | 65.8 | 5.2 | 5.2 | 25.8 | 3.6 | 4.0 |
| | ActIdx-F | 25.8 | 26.7 | 27.9 | 14.6 | 18.6 | 18.7 | 65.6 | 23.6 | 23.6 | 30.3 | 12.6 | 13.3 |
| | ActIdx-B | 25.4 | **31.4** | **31.9** | 17.1 | 28.0 | 28.3 | 66.5 | 30.8 | 32.0 | 29.1 | 21.6 | 22.7 |
| | ActIdx | **27.6** | 30.9 | 31.1 | **17.6** | 29.3 | 29.5 | **66.6** | **32.6** | **33.6** | **31.9** | **24.4** | **25.7** |
| Q-3B | InsOnly | 15.9 | 3.7 | 4.1 | 9.2 | 0.6 | 0.6 | 61.2 | 0.0 | 0.0 | 15.2 | 0.2 | 0.2 |
| | PassIdx | 16.5 | 17.1 | 17.4 | 11.7 | 7.1 | 7.2 | 67.0 | 1.1 | 1.6 | 22.8 | 1.8 | 2.4 |
| | Repeat | 18.6 | 16.4 | 16.5 | 12.2 | 9.1 | 9.3 | 64.0 | 0.9 | 1.2 | 23.1 | 1.9 | 2.2 |
| | Repeat+ | 16.5 | 17.1 | 17.4 | 10.1 | 9.4 | 9.8 | 65.7 | 1.1 | 1.3 | 23.5 | 2.0 | 2.6 |
| | ActIdx-F | 19.8 | 22.6 | 23.1 | 12.5 | 12.8 | 13.3 | 67.7 | 3.0 | 3.0 | **24.7** | 3.9 | 4.7 |
| | ActIdx-B | 19.7 | **24.5** | **24.9** | 14.1 | 19.0 | 19.6 | 65.2 | 17.6 | **23.4** | 24.4 | 7.8 | 14.1 |
| | ActIdx | **21.4** | 23.9 | 24.2 | **15.8** | **19.7** | **19.8** | **67.9** | **20.0** | 23.0 | 24.5 | **10.5** | **15.8** |
| GPT-4.1 | few-shot | 52.7 | 23.0 | 24.0 | 29.6 | 0.0 | 0.0 | 93.0 | 0.0 | 0.0 | - | - | - |

and citation precision. All methods follow a two-stage process: continual pretraining followed by instruction tuning. They differ only in their approach to continual pretraining. The Instruction-only method skips continual pretraining and proceeds directly to instruction tuning. We compare:

**Instruction-only:** Serves as a baseline: the model is instruction-tuned without continual pretraining, testing citation based solely on pre-trained knowledge.

**Passive Indexing:** During continual pretraining, document identifiers are appended to 768-token chunks, allowing the model to passively associate facts with document IDs.

**Repetition:** Following Khalifa et al. (2024), document IDs are appended after each fact to support fine-grained attribution, though frequent insertions may reduce fluency.

**Repetition+:** Continual pretraining appends document identifiers to full documents and sampled segments (e.g., one-third, paragraphs, or sentences) to balance attribution and fluency.

**Active Indexing (Forward):** Intra-document augmentation is used during continual pretraining to reinforce source-to-fact mappings, enhancing grounding within individual documents. This method uses 1.28B augmented tokens (3.3× the original corpus).

**Active Indexing (Backward):** Continual pretraining uses cross-document augmentation to teach fact-to-source mappings, enabling information synthesis and adding 1.47B augmented tokens (3.8×).

**Active Indexing:** Combines forward and backward augmentation, resulting in augmented 2.75B tokens (7.05× original 390M tokens).

While continual pretraining introduces a one-time overhead, our approach adds no inference-time cost. In contrast, the RAG-based method of Gao et al. (2023b) incurs recurring retrieval and conditioning overhead, requiring about 130× more input tokens per query. This trade-off is increasingly attractive in today's *data*-limited regime: high-quality human data is scarce and pure compute scaling shows diminishing returns Villalobos et al. (2024); Muennighoff et al. (2023). Synthetic data is therefore a practical path to new capabilities Qin et al. (2025), making a one-time training investment for scalable citation well justified. See training details in Appendix D. We also evaluate GPT-4.1 with 3-shot citation prompting to measure a proprietary LLM internal citation (note: its decoding space cannot be constrained).

## 5.2 MAIN RESULTS

**Passive Indexing Is Insufficient for Citation.** Table 1 shows that Passive Indexing—simply appending document IDs and expecting implicit learning—is insufficient. While it offers modest gains over instruction tuning alone (e.g., 2.4 vs. 0.9 on RepliQA), citation precision remains low. Attaching doc-IDs closer to facts (Repeat) also fails to improve performance on realistic tasks, unlike

prior work (Khalifa et al., 2024), due to their use of synthetic, rigidly structured data. In real-world settings, diverse and loosely aligned facts limit the effectiveness of sentence-level associations. Nonetheless, Repeat+ performs slightly better, likely due to its more granular ID attachment across different parts of the document, which enhances fluency and supports better generalization.

**Active Indexing is Effective in Both Directions.** Active Indexing consistently outperforms passive baselines across all tasks (Table 1). Backward indexing is stronger than forward, and combining both delivers the largest gains (e.g., citation precision: 2.4→32.6), underscoring their complementarity. We expect further scaling of the augmentation to yield additional improvements (§5.3), and we verify the additive benefits of using both directions in Appendix B.2. Notably, Active Indexing also boosts answer correctness, likely due to exposure to factual content in diverse formats (Allen-Zhu & Li, 2025; Yang et al., 2025). **Model Size Matters.** On Qwen-2.5-3B, we see the same trends as 7B but with much lower citation performance. Scaling model size (Qwen-2.5: 3B→7B→14B; Llama-3: 3B→8B) yields monotonic gains under Active Indexing (Appendix B), suggesting larger models generally support stronger citation when trained for it. Yet model size alone is insufficient: without targeted training (e.g., Active Indexing), even GPT-4.1 struggles with internal citation. Beyond Wikipedia, sources often either lack semantically guessable titles (e.g., arXiv) or lack stable document identifiers altogether (e.g., Common Crawl for ELI5), making internal citation difficult without an explicit indexing objective.

**Pretrained Models Memorize Wikipedia Titles.** On ASQA (Wikipedia-based task), models achieve competitive citation performance even without indexing (e.g., 20.0 with instruction-tuning only vs. 24.1 with passive indexing). We attribute this to two main factors: (1) large models memorize not just verbatim text but also Wikipedia titles during pretraining (Weller et al., 2024; Zhang et al., 2025b), and (2) Wikipedia titles are typically entities, making it easier for models to "shortcut" by predicting the entity as the title. Preliminary results on other Wikipedia-based tasks, such as TriviaQA and HotpotQA, show similar patterns. See Appendix F.2 for details. See Appendix G for qualitative analysis and Appendix B.4 for OOD evaluation analysis for Active Indexing.

## 5.3 WHY DOES ACTIVE INDEXING WORK?

**Facts Variation and Active Supervision Are Both Crucial for Reliable Citations** To identify what drives reliable internal citation, we study how fact variation (i.e., presenting the same facts in different ways) and active supervision affect model performance. Using the RepliQA dataset, we scale the amount of augmented data and evaluate how citation performance varies with the augmentation ratio (Figure 2). We compare three approaches: (1) Passive Indexing with Replay, which simply repeats the same facts and document identifiers with more epochs; (2) Passive Indexing with Synthetic Continual Pretraining (Yang et al., 2025) (PI-SCP), which paraphrases facts and introduces relational variants, but still lacks explicit QA-style training to use document IDs. and (3) Active Indexing, which not only diversifies fact formulations but also explicitly trains the model to use document identifiers in QA-style contexts. Our findings are threefold: 1. **Fact Variation Helps**: Citation performance improves as Active Indexing increases both the scale and diversity of fact pre-

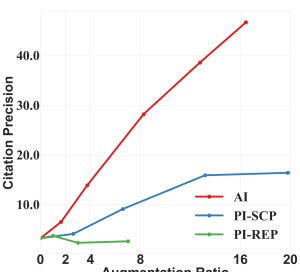

Figure 2: Scaling Comparison Between Active Indexing and Passive Indexing on RepliQA.

sentations. In contrast, simple token replay provides no benefit—and can even degrade performance due to overfitting. 2. **Active Supervision is Crucial:** Although PI-SCP introduces diverse phrasings of the same facts, it still lags behind active indexing. This indicates that diversity alone is insufficient—explicitly teaching the model to use document identifiers in context is essential for robust citation. 3. **Room to grow**: Citation performance continues to improve at the highest augmentation levels tested (16×), suggesting that further gains are possible beyond our current limits.

**Active Indexing Learns Beyond Semantic Shortcuts** A key concern is that Active Indexing might rely on *semantic shortcuts* from titles (i.e., "guessing the title") instead of learning true fact→ID associations. We test this shortcut hypothesis on the RepliQA-only setup by measuring how semantically distinctive each title is. For each RepliQA statement (query + answer), we compute its semantic similarity (sentence embeddings) to all document titles and rank the true title among

them. We treat this rank as *Title Distinctiveness*: Easy = true title is highly similar to the statement (semantic shortcuts possible); Hard = many other titles are more similar (semantic cues unreliable). We bucket examples into four bins (Easy–Very Hard) and report citation precision and average rank in Table 2. Findings are: 1. **Titles are not semantically unique.** In >90% of cases, at least one other title is more similar to the statement than the true title; on average, the correct title ranks 208th out of 6,822. Simple "nearest-title" guessing is rarely sufficient. 2. **Semantic cues help when available.** C-Pre is higher in the Easy bin and decreases as titles become less distinctive, showing the model sensibly uses semantic cues when they exist. 3. **Non-trivial performance without shortcuts.** Even in the Hard/Very Hard bins, where semantic similarity is a poor signal, the model still achieves ∼40% C-Pre. This indicates Active Indexing learns genuine fact→ID associations, not just shortcuts.

Table 2: **Citation precision vs. title distinctiveness on RepliQA.** Active Indexing benefits from distinctive titles but maintains substantial accuracy even when titles are not semantically similar, indicating learning beyond semantic shortcuts.

|  | Easy | Medium | Hard | Very Hard | Total |
|---|---|---|---|---|---|
| C-Pre | 55.9 | 49.6 | 40.1 | 40.0 | 46.7 |
| Avg. Rank | 2 | 10 | 60 | 761 | 208 |

**Bridging Memorization and Generalization in Citation** Past work has shown that stronger memorization does not guarantee better generalization (Wang et al., 2025b). We observe a similar pattern in citation tasks. There is a crucial difference between how LLMs (1) **memorize** document identifiers during continual pretraining, and (2) **generalize** to use these identifiers when answering downstream questions. To probe this, we evaluate four setups that progressively shift from pure memorization to downstream usage: 1. **FullDoc:** Predicting the doc-ID given the full document. 2. **PartialDoc:** Predicting the doc-ID from a partial document segment. 3. **GoldQA:** Predicting the doc-ID given the question and gold answer. 4. **ModelQA:** Predicting the doc-ID given the question and the model's generated answer. We reuse RepliQA 7B models and report Hit@1/10 accuracy. As shown in Table 3, all methods' performance declines as tasks shift from memorization to generalization in downstream citations. Notably, more replay epochs (PassIdx-REP) improve memorization (FullDoc: 27.0 → 74.6) but hurt downstream generalization (ModelQA: 7.8 → 6.0), suggesting overfitting to shallow patterns. In contrast, Active Indexing bridges this gap by promoting both memorization and effective usage in QA, encouraging robust fact-to-source grounding.

Table 3: Document ID Memorization and Generalization.

|  | FullDoc | | PartialDoc | | GoldQA | | ModelQA | |
|---|---|---|---|---|---|---|---|---|
|  | Acc@1 | Acc@10 | Acc@1 | Acc@10 | Acc@1 | Acc@10 | Acc@1 | Acc@10 |
| PassIdx | 27.0 | 80.4 | 5.8 | 31.8 | 8.6 | 21.4 | 7.8 | 19.6 |
| PassIdx-REP | 74.6 | 94.4 | 10.6 | 32.8 | 6.6 | 25.6 | 6.0 | 22.4 |
| ActIdx | 95.2 | 100.0 | 72.8 | 97.4 | 66.4 | 94.2 | 54.2 | 88.4 |

## 5.4 INTERNAL VS. EXTERNAL CITATIONS

We compare internal and external citation methods to understand their respective limitations and the potential for synergy. External citations allow models to access information beyond their memorized knowledge, while internal citations serve as a robust fallback when retrieval fails. This trade-off becomes especially important when retrieval quality varies. To explore this, we compare:
**Internal Only**: Generate internal citations without retrieval using Active Index.
**External Only**: Generate citations via RAG with 3-shot examples and top-5 retrieved documents, following the best-performing setup in Gao et al. (2023b).
**Hybrid Joint**: Instruction-tuning ActiveIdx to consume retrieved documents, generating both internal and external citations after an initial assessment of document sufficiency during inference.
**Hybrid Fallback**: A pipeline that first attempts RAG, falling back to Hybrid Joint if retrieved documents are deemed insufficient by the model.

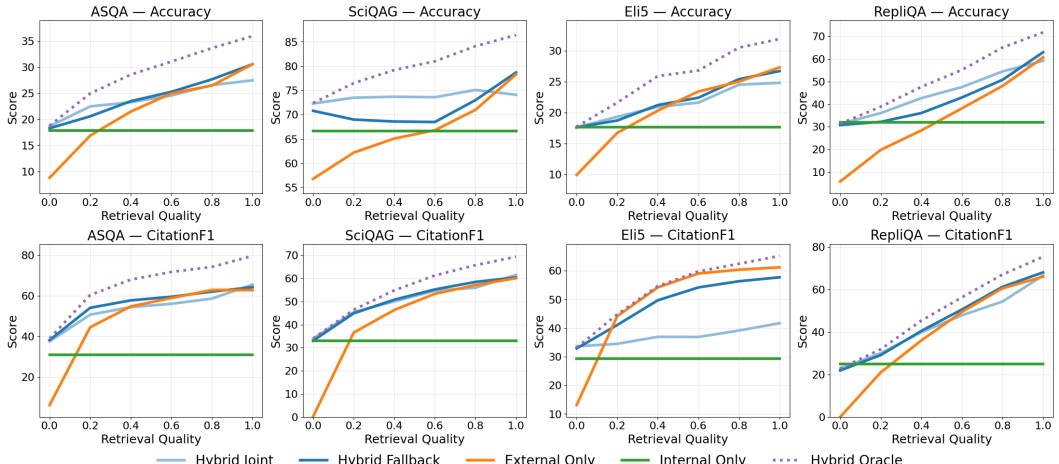

Figure 3: Performance of internal, external, and hybrid citations across retrieval quality (0=sparse retrieval, 1=dense retrieval). Internal only excels under poor retrieval, external only under strong retrieval, while hybrids generally perform best regardless of retrieval quality, with room to improve.

**Hybrid Oracle**: A conceptual upper-bound that selects the better output from the Internal and External Only methods for each example. See more experiments details in Appendix D.2.

We test these strategies across a spectrum of retrieval quality, which we simulate by interpolating between sparse retrieval (BM25; lower-quality in our setup) and dense retrieval (Lin et al., 2021; high-quality). Retrieval quality = 0 uses BM25's top-5; quality = 1 uses dense top-5; intermediate values (e.g., 0.2) mix the two proportionally (20% sampled from dense, 80% BM25). As shown in Figure 3, when retrieval is poor, methods incorporating internal citations (Internal Only and Hybrid variants) significantly outperform External Only, highlighting the importance of internal fallback under noisy retrieval. As retrieval improves, external outperforms internal, showing that reliable external evidence can complement internal knowledge. Hybrid Approach, taking advantages of both, perform generally best regardless of retrieval quality. Notably, a performance gap remains between our best Hybrid method and the Hybrid Oracle, pointing to headroom for more effective integration strategies that better reconcile retrieved and memorized knowledge. We also provide a finer-grained analysis across conflict slices, partitioning examples by whether the internal-only and external-only systems are correct (Appendix B.3).

## 6 CONCLUSION

We show that large language models can attribute answers to their pretraining data without relying on test-time retrieval. We (1) introduce *CitePretrainBench* for internal citation across short- and long-form QA, and (2) propose *Active Indexing*, a continual pretraining strategy that teaches models to link content with document identifiers. Key findings include: 1. **Teaching beats hoping**: Active Indexing, which frames citation as real tasks, improves precision and recall by up to 32 points—outperforming passive approaches. 2. **Complementary directions**: Forward and backward augmentations are most effective when combined. 3. **Scale helps**: Performance improves consistently with more augmented data. 4. **Internal and external are complementary**: Combining both enhances robustness to retrieval quality. See Appendix I for limitations and future directions.

### ACKNOWLEDGMENTS

Yukun Huang, Sanxing Chen, and Bhuwan Dhingra's research is supported in part by the NSF award IIS-2211526. Jian Pei's research is supported in part by the NSF DMS-2434666. All opinions, findings, conclusions and recommendations in this paper are those of the authors and do not necessarily reflect the views of the funding agencies.

## REPRODUCIBILITY

We are committed to ensuring the reproducibility of our results. We describe details of our training procedures (§4,) and implementations (§5.1, Appendix D). For our evaluation, we describe details of datasets (§3.2), metrics (§3.3, §3.4) and processing steps (Appendix A). Code and data are released in github.

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

# A  DATASETS

## A.1  CORPUS CONSTRUCTION

**Common Crawl**  To construct the essential document set, we require full documents for continual pretraining—unlike the Sphere corpus (Piktus et al., 2021), which consists of 100-word passages. We process raw 2019 Aug Common Crawl snapshots using CCNet (Wenzek et al., 2020) to obtain clean full-text documents. To identify documents relevant to ELI5, we first index 100-word passages (in Sphere format) and use BM25 to retrieve the top-100 passages for each question in the ELI5 train, dev, and test sets. We then use an LLM to decompose each question into self-contained fact claims, treating each claim as a separate query to retrieve the top-200 relevant passages.

Next, we verify whether the retrieved passages support each claim and compile a support set. This set is combined with the original relevant passages, and using passage metadata, we locate and extract the corresponding full documents from Common Crawl. This process yields our core document set, comprising 30,025 documents with a total of 110,594,287 tokens, averaging 3,683 tokens per document. For training set instances with unsupported claims, we use GPT-4o to generate synthetic documents styled like Common Crawl, ensuring the model is not trained on unsupported inputs. This results in a corpus from the **Other** source, containing 7,593 documents with 2,210,346 total tokens, averaging 291 tokens per document.

The data is licensed under CC0.

**Wikipedia**  We build on the KILT knowledge source (Petroni et al., 2021), based on the 2019/08/01 Wikipedia snapshot. We include source documents from ASQA, the TriviaQA dev set, and a subset of HotpotQA (medium-level questions), mapped to KILT IDs. This forms our essential Wikipedia corpus, consisting of 30,025 documents totaling 110.6M tokens (avg. 3,683 tokens/document).

The data is licensed under CC BY-SA 3.0.

**ArXiv**  SciQAG (Wan et al., 2024) provides scientific papers without titles. We retrieve titles using the papers' DOIs from arXiv, resulting in 22,743 documents with 114.0M tokens (avg. 5,013 tokens/document).

arXiv metadata is used under the CC0; license for each paper varies.

**RepliQA**  We use the first two splits of the RepliQA dataset (Monteiro et al., 2024) as the full corpus, yielding 7,182 documents and 8.88M tokens (avg. 1,236 tokens/document). Due to frequent title duplication, we use an LLM to generate more descriptive titles.

This core corpus totals approximately 392M tokens. The data is licensed under CC BY 4.0.

## A.2  RENAME DUPLICATED TITLES

Due to frequent title duplication—especially in Common Crawl and RepliQA—we adopt a renaming strategy using an LLM. For each duplicated title, we iteratively rename the document until all titles are unique. We also perform cross-source deduplication after renaming.

Table 4: Full results for INSONLY, PASSIDX, and ACTIDX across all five base models on four QA datasets. Acc = answer correctness, C-Pr = citation precision, C-Re = citation recall.

| Model | Method | ASQA | | | Eli5 | | | SciQAG | | | RepliQA | | |
|---|---|---|---|---|---|---|---|---|---|---|---|---|---|
| | | Acc | C-Pr | C-Re | Acc | C-Pr | C-Re | Acc | C-Pr | C-Re | Acc | C-Pr | C-Re |
| Q-3B | InsOnly | 15.9 | 3.7 | 4.1 | 9.2 | 0.6 | 0.6 | 61.2 | 0.0 | 0.0 | 15.2 | 0.2 | 0.2 |
| | PassIdx | 16.5 | 17.1 | 17.4 | 11.7 | 7.1 | 7.2 | 67.0 | 1.1 | 1.6 | 22.8 | 1.8 | 2.4 |
| | ActIdx | 21.4 | 23.9 | 24.2 | 15.8 | 19.7 | 19.8 | 67.9 | 20.0 | 23.0 | 24.5 | 10.5 | 15.8 |
| Q-7B | InsOnly | 19.1 | 20.0 | 21.2 | 11.5 | 5.9 | 6.4 | 65.9 | 0.6 | 0.8 | 24.2 | 0.9 | 1.3 |
| | PassIdx | 21.5 | 24.1 | 24.2 | 14.5 | 8.9 | 9.0 | 65.7 | 2.4 | 2.4 | 24.8 | 2.4 | 2.5 |
| | ActIdx | 27.6 | 30.9 | 31.1 | 17.6 | 29.3 | 29.5 | 66.6 | 32.6 | 33.6 | 31.9 | 24.4 | 25.7 |
| Q-14B | InsOnly | 26.4 | 25.1 | 27.5 | 14.9 | 5.9 | 6.3 | 67.4 | 0.5 | 0.8 | 27.6 | 0.75 | 0.9 |
| | PassIdx | 30.0 | 27.1 | 27.3 | 18.5 | 9.3 | 9.4 | 67.4 | 2.3 | 2.3 | 28.4 | 2.0 | 2.0 |
| | ActIdx | 31.6 | 40.9 | 41.3 | 19.3 | 33.2 | 33.6 | 70.2 | 34.8 | 35.1 | 37.0 | 30.8 | 31.5 |
| L-3B | InsOnly | 18.0 | 11.9 | 18.1 | 9.4 | 3.6 | 3.9 | 61.8 | 0.2 | 0.2 | 19.0 | 0.27 | 0.4 |
| | PassIdx | 21.5 | 18.5 | 19.1 | 11.2 | 7.1 | 7.9 | 63.2 | 1.8 | 2.1 | 23.4 | 1.8 | 2.0 |
| | ActIdx | 25.9 | 29.0 | 29.3 | 15.6 | 25.9 | 26.3 | 65.6 | 13.2 | 32.8 | 26.8 | 13.9 | 23.0 |
| L-8B | InsOnly | 25.4 | 23.3 | 25.5 | 12.1 | 5.0 | 5.5 | 67.3 | 1.0 | 1.6 | 24.1 | 0.7 | 1.4 |
| | PassIdx | 28.6 | 27.8 | 28.1 | 16.1 | 9.3 | 9.5 | 65.1 | 3.6 | 3.7 | 25.6 | 1.9 | 1.9 |
| | ActIdx | 28.2 | 31.7 | 31.8 | 17.9 | 30.7 | 30.8 | 67.7 | 39.3 | 40.3 | 35.7 | 36.2 | 36.9 |

## A.3 DOWNSTREAM TASKS

For ELI5, the test set is sourced from (Gao et al., 2023b), while the training and development sets are derived from a newer version.[3] Any duplicates from the test set have been removed from the training and development sets. ELI5 is licensed under BSD 3.

For ASQA, we map annotated source documents to the KILT knowledge base and filter out datapoints with unmatched sources, resulting in 863 test examples. ASQA is licensed under CC BY 4.0.

For SciQAG, due to noise in the training set, we re-split the original test set into train, dev, and test splits based on documents, ensuring no title overlap. This yields 853, 95, and 300 documents, respectively. Evaluation is performed on 1,000 sampled questions from the 300 test documents. SciQAG is released on GitHub without explicit license, we will not redistribute it without author acknowledgment.

For RepliQA, we similarly split train, dev, and test sets by documents. The test set includes 1,000 QA pairs sampled from the test documents. RepliQA is licensed under CC BY 4.0.

## A.4 INSTRUCTION TUNING SET

We randomly sample 1,000 training examples from ASQA, ELI5, RepliQA, SciQAG, and HotpotQA (medium level), and 200 from each dev split to form a validation set for early stopping during instruction tuning.

# B ADDITIONAL RESULTS

## B.1 MAIN RESULTS OF MORE MODELS

We also run active-indexing experiments on Qwen-2.5-14B, Llama-3.2-3B, and Llama-3.1-8B. Performance increases monotonically as model capacity scales—Qwen from 3B → 7B → 14B and Llama from 3B → 8B—further confirming that citation ability improves with model size. As shown in Table 4, Active Indexing consistently outperforms Passive Indexing across models and datasets. Also, as the model size increases, the performance of Active Indexing improves monotonically.

---

[3]https://huggingface.co/datasets/rexarski/eli5_category

## B.2 Complementarity of forward and backward

We study scaled complementarity of forward and backward scale effect on RepliQA. We augment the RepliQA corpus: 66M tokens from Active Indexing Forward (AIF), 76M from Backward (AIB), and their combination (AI), totaling 16.5× the original size. From these, we subsample different proportions to construct a scaling curve. Figure 4 shows citation precision on RepliQA (y-axis) versus the fraction of augmented data used (x-axis). Key findings include: 1. **Scaling helps:** All variants—AIF, AIB, and AI—improve with more augmented data, even up to 16.5×. 2. **Complementary directions:** AIF alone is less effective, but combining AIF with AIB (i.e., AI) consistently outperforms either alone. Each point on the AI curve uses matched subsets of AIF and AIB, indicating additive and reinforcing effects. 3. **The role of forward** While backward provides most gains, but forward provides consistent additive effect, suitable for users willing to trade-off compute for better performance. Moreover, It supports answering realistic queries like "Explain document X / paper Y" in a closed-book manner, which requires the model to internally condition on document identifiers.

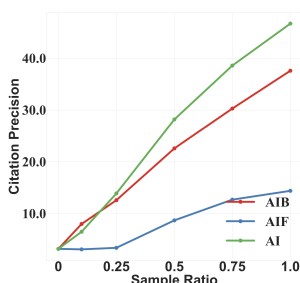

Figure 4: Scaling Curve of Combining Backward and Forward on RepliQA

## B.3 Behavior of Hybrid Systems under Internal–External Conflicts

Hybrid systems are most useful in the conflict regime, where the model's internal belief disagrees with the retrieved context. To make this behavior explicit, we analyze RepliQA by partitioning examples into three slices based on correctness of the internal-only and external-only systems: 1. Internal wrong / External correct (Int=0, Ext=1), 2. Internal correct / External wrong (Int=1, Ext=0), and 3. No conflict (both correct or both wrong).

Table 5 reports both (i) the proportion of each slice and (ii) the accuracy of Internal-only, External-only, Hybrid-Joint, Hybrid-Fallback, and Hybrid-Oracle in each slice, under high-quality retrieval (retrieval quality = 1.0) and noisy retrieval (0.2).

Under high-quality retrieval (1.0), the dominant conflict regime is precisely the "production reality" where retrieval corrects the model: the Int=0, Ext=1 slice accounts for 38.6% of examples, compared to only 8.3% for Int=1, Ext=0. In this regime, Hybrid-Fallback behaves almost like an ideal "defer-to-retrieval" policy: it nearly matches External-only accuracy in the Int=0, Ext=1 slice (99.5 vs. 100.0), while Internal-only is always wrong by construction (0.0). Hybrid-Joint is more conservative: it substantially improves over Internal-only (75.6 vs. 0.0) but falls short of External-only, reflecting its stronger reliance on internal knowledge even when retrieval is correct.

When retrieval quality degrades (0.2), the picture flips: the Int=1, Ext=0 slice becomes more prevalent (22.0% vs. 9.9% for Int=0, Ext=1). In this setting, both hybrid methods shift their arbitration toward internal knowledge. For example, in the Int=1, Ext=0 slice, Hybrid-Joint attains 58.2 accuracy, substantially outperforming External-only (0.0) and improving over Internal-only's baseline, while Hybrid-Fallback also achieves strong gains relative to External-only. This demonstrates that the hybrids do not blindly trust retrieval; they adaptively fall back to parametric knowledge when the retrieved context is unreliable.

Finally, even in the no-conflict slice where both Internal-only and External-only are wrong, Hybrid-Joint can outperform both by aggregating partial evidence from the two sources. This reflects a second layer of complementarity beyond simple source selection: hybrid reasoning can construct better answers by jointly leveraging noisy parametric and contextual signals, not just by choosing between them.

Overall, this analysis directly quantifies the conflict slices and shows that our hybrid strategies implement sensible arbitration policies: when retrieval is reliable, they behave like "switch-to-external" systems; when retrieval is noisy, they lean back on internal knowledge while still extracting value from combining both signals.

| RQ | Method | Int wrong / Ext correct | Int correct / Ext wrong | No conflict | Total |
|----|--------|------------------------|------------------------|-------------|-------|
| | Hybrid-Joint | 75.6 | 51.8 | 48.4 | 59.2 |
| | Hybrid-Fallback | 99.5 | 6.0 | 44.4 | 62.5 |
| 1.0 | Hybrid-Oracle | 100.0 | 100.0 | 44.4 | 70.5 |
| | External-only | 100.0 | 0.0 | 44.4 | 62.2 |
| | Internal-only | 0.0 | 100.0 | 44.4 | 31.9 |
| | Hybrid-Joint | 66.7 | 58.2 | 24.5 | 36.1 |
| | Hybrid-Fallback | 88.9 | 41.8 | 21.9 | 32.9 |
| 0.2 | Hybrid-Oracle | 100.0 | 100.0 | 14.5 | 41.8 |
| | External-only | 100.0 | 0.0 | 14.5 | 19.8 |
| | Internal-only | 0.0 | 100.0 | 14.5 | 31.9 |

Table 5: Behavior of hybrid systems under internal–external conflicts on RepliQA. Entries are accuracies within each correctness slice: Internal wrong / External correct, Internal correct / External wrong, and No conflict. For retrieval quality 1.0, the slice proportions are 38.6%, 8.3%, and 53.1%, respectively; for retrieval quality 0.2, they are 9.9%, 22.0%, and 68.1%.

### B.4 OUT-OF-DOMAIN EVALUATION AFTER ACTIVE INDEXING

We evaluate the out-of-domain behavior of models after *RepliQA-only* continual pre-training (CPT). We measure (i) perplexity on held-out Wikipedia and ArXiv corpora, and (ii) TriviaQA accuracy after task-specific fine-tuning. Consistent with Khalifa et al. (2024), we observe an increase in perplexity on natural text after CPT. We attribute this rise to two factors: (1) **Domain-specific CPT**, which is known to induce catastrophic forgetting when the CPT corpus is significantly narrower than the original pre-training distribution; and (2) **Document identifiers**, whose form (e.g., natural-language titles vs. integers) introduces different levels of distributional shift.

To disentangle these effects, we train five CPT variants on the same backbone with identical token budgets: (1) raw data (no titles), (2) Passive Indexing with natural-language title IDs, (3) Passive Indexing with integer IDs, (4) Passive Indexing with repeated natural-language IDs, and (5) Active Indexing. We then evaluate all models using the same OOD metrics.

| CPT Method | Wiki PPL | ArXiv PPL | TriviaQA Acc. |
|-----------|----------|-----------|---------------|
| Base Model (No CPT) | 6.71 | 4.61 | 56.0 |
| Raw Data (No Title) | 32.15 | 6.65 | 50.8 |
| Passive Index (Title ID) | 54.31 | 6.90 | 49.5 |
| Passive Index (Integer ID) | 60.15 | 6.88 | 49.2 |
| Passive Index Repeat (Title) | 39.78 | 6.80 | 51.2 |
| Active Index | 26.52 | 5.40 | 50.5 |

Table 6: OOD evaluation after RepliQA-only continual pre-training. Lower perplexity is better. Higher accuracy on triviaqa is better.

**Findings.** **1. All CPT variants raise OOD perplexity**, confirming that narrow-domain CPT is the dominant source of forgetting, as even the Raw Data (no-title) model shows substantial degradation. **2. Natural-language IDs are less harmful than integer IDs**: among Passive Index variants, perplexity follows the pattern natural-text < repeated-natural-text < integer-ID, indicating that well-formed textual identifiers integrate more smoothly into language modeling. **3. Repeated titles are not particularly damaging**: unlike Khalifa et al. (2024), we do not observe large penalties from repetition, likely because our identifiers are natural text rather than opaque tokens. **4. Active Indexing is the least harmful variant**: it obtains the lowest perplexity on both Wikipedia and ArXiv, suggesting its augmentation procedure improves linguistic fluency within this knowledge domain. **5. OOD effects vary across corpora**: perplexity increases more on Wikipedia than on ArXiv, indicating that forgetting depends on similarity between the CPT distribution and the target corpus. Finally, **6. perplexity does not correlate with QA performance**: despite substantial perplexity differences across variants, TriviaQA accuracy after fine-tuning remains similar, implying that perplexity shifts reflect stylistic or distributional drift rather than loss of underlying knowledge.

## C    MORE RELATED WORK

**More External Citations**    A parallel line of external citation work targets *finer-grained* grounding, at the sentence/span and even token level, primarily in open-book settings. Systems such as *GopherCite* explicitly interleave answers with short, verified quotes and learn to abstain when unsure, illustrating span-level attribution tied to retrieved evidence. (Menick et al., 2022) Semi-extractive generation (*SEMQA*/QuoteSum) enforces copy-and-connect outputs, yielding inline quoted spans by construction, (Schuster et al., 2024) while *locally-attributable* generation optimizes for concise, sentence-local citations consumers can check quickly (Slobodkin et al., 2024). For long-context QA, *LongCite* and its LongBench-Cite benchmark train models that produce answers with *sentence-level* citations in one pass (Zhang et al., 2025a). They still cite from the external context and can't produce citations in a closed-book manner. On the evaluation side, *EXPERTQA* contributes expert-curated questions and *expert-verified, claim–evidence* annotations for long-form answers—auditing system-provided citations and enabling claim-level scoring in high-stakes domains. (Malaviya et al., 2024) Methods that unify retrieval and reflection (e.g., *Self-RAG*), add self-critique flags to segment-level citations. (Asai et al., 2024) Beyond during-generation approaches, *post-hoc* attribution retrofits support after the fact. *RARR* ("research & revise") finds evidence and edits outputs to align claims with sources, (Gao et al., 2023a) and follow-ups for long documents decompose answers into factual units before mapping each unit to supporting sentences, improving coverage of fine-grained support. (Sancheti et al., 2024; Ramu et al., 2024) Finally, token-level context credit assignment (e.g., *Token-Shapley*) scores which specific context tokens support each generated token. (Xiao et al., 2025) These directions still rely on external evidence at inference, and thus constitute *external* citation.

**Data attribution**    seeks the training examples that most influenced a model's behaviour (Park et al., 2023) (e.g., via influence functions or gradient tracing), while **fact attribution** asks which facts support a given answer—often handled by external retrieval or by pretraining identifiers that may not coincide with truly influential data. Recent analysis-time tools underscore this gap: *OLMoTrace* traces verbatim spans in model outputs back to specific pretraining documents across multi-trillion-token corpora in real time, and *RapidIn* retrieves token-wise influential training points at scale via compressed gradient caching (Liu et al., 2025; Lin et al., 2024). Empirical studies further show that examples ranked as most *influential* need not be those that explicitly *contain* the cited fact, and that alignment between the two increases with model/corpus scale (Chang et al., 2025). These objectives therefore do not necessarily align. **Active Indexing bridges them**: by presenting each training fact with its document identifier in a QA-style format, it explicitly couples the influential data point with its provenance, so that reproducing an identifier at inference time simultaneously evidences both data lineage and factual entailment—achieving traceable citations without expensive post-hoc analysis and aligning factual correctness with the model's true training history.

**Copyright, Transparency, and Legal Considerations**    The legal status of training on copyrighted content is the subject of active policy debate. (Chen et al., 2024) By encouraging models to surface explicit provenance, our approach offers a technical step toward such transparency: citations make it easier to audit whether a model relies on protected material. However, we also highlight a dual risk in Appendix I: stronger provenance can *increase* exposure of private or proprietary text snippets. Exploring privacy-preserving identifiers (e.g., hashed IDs, differential privacy) and selective redaction during generation remains important future work.

## D    IMPLEMENTATION DETAILS

### D.1    TRAINING

We perform continual pretraining using 4 H200 GPUs for 3 epochs across all methods in the main experiments, with a batch size of 256, a maximum context length of 2048, and a constant learning rate of 5e-5 with 10 warm-up steps. With 8-bit AdamW paged optimizers, each model fits on a single GPU and can be trained using DDP without requiring FSDP. For instruction tuning, we use a linear decay scheduler with a learning rate of 5e-6, train for up to 5 epochs with early stopping, and use a batch size of 64.

The longest continual pretraining run (Qwen-2.5-7B on 3B tokens × 3 epochs = 9B tokens) takes 320 H200 GPU hours (80h × 4 GPUs). The shortest on 7B (Passive Indexing with 400M × 3 = 1.2B tokens) takes 43 H200 GPU hours.

## D.2 INTERNAL CITATIONS V.S. EXTERNAL CITATIONS

In the experiments of comparing internal citation with external citations, we vary retrieval quality from sparse (BM25) to dense retrieval (Lin et al., 2021) by mixing top-5 documents retrieved by each method. Specifically, we construct hybrid top-5 sets by randomly sampling a proportion of documents from the sparse and dense retrieval outputs. We report both answer correctness and citation f1 which combines citation precision and recall as a single score.

And more details for our hybrid methods are below:

**Hybrid Joint**: To train a model that can leverage both internal and external knowledge, we fine-tune the Active Indexing model to handle retrieved documents as input. For each question in the instruction tuning set (Appendix A.4), we retrieve external documents using both sparse and dense methods. We then create two training instances per question: One with sparse-retrieved documents + question as input. One with dense-retrieved documents + question as input. In both cases, the target output is the correct answer with citations. This trains the model to produce grounded answers regardless of the quality of the provided documents. We find performance improves when we adopt a chain-of-thought format, where the model first reflects on document sufficiency before answering, rather than directly generating an answer.

**Hybrid Fallback**: At inference time, we first use a 3-shot prompt in the RAG setup to generate answers using the top-5 retrieved documents. The prompt allows the model to abstain if it finds the documents insufficient. If it abstains, we fall back to the Hybrid Joint model, which can rely on both external and internal knowledge. If the documents are deemed sufficient, we proceed with standard external-only RAG generation with citations.

# E CITATIONS EVALUATION

## E.1 LONG-FORM CITATIONS EVALUATION

For each model-generated long-form answer, we first use GPT-4.1 to decompose it into self-contained claims, each linked to its cited source. We then retrieve the corresponding documents from a MongoDB-based [4] corpus. Since documents are often too long for the NLI model's (Gekhman et al., 2023) input, we chunk each into 512-token segments and retrieve the most relevant ones. A citation is considered correct if any chunk entails the claim. Citation precision is defined as the proportion of citations that support their claims, and recall as the proportion of claims that are fully supported.

## E.2 SHORT-FORM CITATIONS ERROR ANALYSIS

We conduct citation analysis on the best Qwen-2.5-7B model trained on RepliQA with Active Indexing. Using GPT-4.1, we assess whether the cited documents entail the model-generated answers. We find that 7% of examples cite documents that support the answer but do not match the gold citation, indicating that true citation accuracy may be underestimated. However, 6.2% of examples include correct citations but incorrect answers, suggesting that the model may sometimes retrieve the right source while generating an incorrect response.

# F PRELIMINARY EXPERIMENTS

## F.1 DOCUMENT IDENTIFIERS

We investigate which types of document identifiers are most effective for language models to memorize during continual pretraining. Experiments are conducted on the SciQAG dataset using LLaMA-3.1-1B. To evaluate memorization, we measure accuracy@1 and accuracy@10, where the

---

[4]https://www.mongodb.com

model is given the document content and asked to rank possible document IDs by the summed log-probability of each ID. Higher scores indicate stronger memorization of the document-identifier associations. We compare different strategies for constructing document identifiers, evaluating their effectiveness in continual pretraining for memorization.

**1. Natural Title (Raw):** A baseline without continual pretraining. The model is directly prompted to rank text titles given document content. This tests whether pretrained LMs can match content to titles without exposure.

**2. Natural Titles** : We perform continual pretraining where each document is appended with its human-written text title. This approach uses natural-language identifiers that align with the model's training distribution.

**3. Hierarchical K-Means Integer (HKM-Integer):** Instead of using random integers, we construct semantically structured integer IDs following Tay et al. (2022). Documents are embedded and clustered using K-means into 10 top-level groups. Each group is assigned a prefix digit. The process is recursively applied within each cluster, with each level adding a digit to the ID. Documents with shared prefixes are semantically similar, making it easier for the model to generalize over structured identifiers.

**4. Hierarchical LDA with Keyword Labels (HLDA-Keywords)** We apply hierarchical topic modeling (LDA) to recursively cluster documents. For each cluster, we use an LLM to generate a representative keyword based on its most salient documents. The final identifier is a concatenation of these keywords along the cluster path, forming a semantic, hierarchical label.

**5.Domain-First Keyword Identifier (Domain→Keywords)** Each SciQAG document is tagged with a domain and associated keywords. We construct identifiers by concatenating the domain name with its keywords, creating a top-down semantic label (e.g., physics-energy-entropy).

**6. Keyword-First Domain Identifier (Keywords→Domain)** Similar to the above, but constructed in a bottom-up manner. Keywords appear first, followed by the broader domain label (e.g., entropy-energy-physics), emphasizing specificity before generality.

Table 7: Results of different document identifiers on SciQAG.

| ID Type | Acc@1 | Acc@10 |
|---|---|---|
| Natural Titles (Raw) | 9.7 | 46.3 |
| Natural Titles | 53.3 | 75.3 |
| HKM-Integer | 2.0 | 21.7 |
| HLDA-Keywords | 32.0 | 50.7 |
| Domain->keywords | 28.7 | 47.7 |
| keywords->Domain | 26.7 | 45.3 |

As shown in Table 7, even semantically structured integer-based identifiers perform significantly worse than text-based methods. We attribute this to the nature of continual pretraining, where the document is directly followed by its identifier. In this setup, natural text provides a more effective and fluent learning signal for the language model.

Among the text-based methods, natural titles achieve the highest performance. We hypothesize three reasons for this: (1) natural titles tend to capture the most salient information from the document and have high information density, (2) they are more fluent and better aligned with the model's pretraining distribution, and (3) the SciQAG corpus is relatively small—within the memorization capacity of the model—so the benefits of structured or compressed identifiers (like integer codes) are less pronounced. Such structured identifiers may only offer advantages at larger scales where memory constraints become a limiting factor.

### F.2 WIKIPEDIA TASKS

We evaluate our approach on two popular Wikipedia-based QA benchmarks: TriviaQA (short-form) and HotpotQA (long-form). For HotpotQA, we focus on medium-difficulty, two-hop questions to match the capabilities of Llama-3.1-1B. We compare two settings: (1) a Raw model directly instruction-tuned on each task, and (2) Passive Indexing, which adds a continual pretraining stage on the Wikipedia corpus before instruction tuning. For TriviaQA, we use Exact Match for Correctness and Citation Precision for citation quality. While for HotpotQA, we use Exact Match for Correctness and both Citation Precision and Citation Recall for citation quality.

As shown in Table 8, continual pretraining provides no noticeable gains. The strong QA and citation performance of the raw model indicates that LLaMA-3 already memorizes much of Wikipedia's content and titles during pretraining, leaving limited headroom for further improvement via continual pretraining. Interestingly, citation accuracy exceeds QA accuracy, suggesting that the model can often guess the correct title even without fully answering the question—likely because Wikipedia articles are topically coherent and revolve around predictable entities.

Table 8: Preliminary Experiments on TriviaQA and HotpotQA

| 1B | | | TriviaQA | | | HotpotQA | | |
|---|---|---|---|---|---|---|---|---|
| | Corpus | #Tokens | Acc | C-Pr | C-Re | Acc | C-Pr | C-Re |
| Raw | - | - | 34.7 | 59.6 | - | 26.0 | 55.9 | 50.7 |
| PassIndex | Wiki | 418M | 31.3 | 61.4 | - | 25.3 | 54.7 | 51.1 |

## G QUALITATIVE EXAMPLES OF CITE-PRETRAIN BEHAVIOR

We perform a manual inspection of model outputs across both long-form and short-form QA settings, highlighting typical behavior patterns—both strengths and failure modes—alongside concrete examples.

### 1. CORRECT ANSWER WITH FAITHFUL AND DIVERSE CITATIONS

In ideal cases, the model not only produces a factually accurate and coherent answer, but also cites multiple distinct documents, each supporting a different part of the response. This indicates that the model is capable of recalling parametric knowledge and integrating diverse sources during generation.

**Example:**
**Question**: *In 2012 the remains of King Richard III were found in a carpark in Leicester. How can scientists be so certain that they've found him?*
**Model Answer:**

> Scientists identified King Richard III through a combination of archaeological evidence and forensic analysis.
> `<|New Scientist Confirms Richard III Skeleton Find with 99.9994% Probability|>`
> ...a skeleton under a council car park...
> `<|Richard III Burial Site: Uncovering the Significance of the Greyfriars Church Choir|>`
> ...scoliosis in the spine, consistent with historical descriptions...
> `<|Richard III of England|>`
> ...head wounds matched reports of his death...
> `<|Leicester Car Park Sighting: DNA Tests Confirm Richard III's Remains Found...|>`
> ...DNA matched two living relatives...
> `<|New Scientist Confirms Richard III Skeleton Find with 99.9994% Probability|>`
> Citation Precision

**Correctness Score:** 0.667 **Citation Score:** 1.0

The answer addresses the question with high fidelity, and each citation grounds a different factual element. This demonstrates the model's ability to attribute evidence at a fine-grained level across documents.

### 2. CORRECT ANSWER BUT INCORRECT CITATIONS

In some cases, the generated answer is factually correct and well-structured, but the cited documents are irrelevant. This suggests a mismatch between content planning and citation generation.

**Example:**
**Question**: *Why do online communities crumble as they gain popularity?*
**Model Answer:**

```
Communities may lose cohesion as they scale.
<|Understanding the Far-Right in Canada:  A
Montreal-Antifasciste Perspective|>
...larger groups face moderation and trust challenges...
<|Understanding the Far-Right in Canada:  A
Montreal-Antifasciste Perspective|>
...trolls and bad actors can increase toxicity...
<|How Reddit's New Approach Tackles Hate Speech without
Banning Communities | WIRED|>
...member attrition often follows hostile interactions...
<|Navigating Online Privacy:  Protecting Your Rights in the
Digital Age|>
...balancing inclusivity with safety becomes harder...
<|Understanding the Far-Right in Canada:  A
Montreal-Antifasciste Perspective|>
```

**Correctness Score:** 1.0 **Citation Score:** 0.0

The answer is broadly accurate and well-articulated, but the cited documents bear no meaningful relation to the explanation. The model appears to attribute wrongly.

### 3. FAITHFUL CITATIONS BUT INCOMPLETE ANSWER

Sometimes, the model successfully grounds all claims in real documents, but the final answer fails to directly address the question.

**Example:**
**Question**: *Why do so many drugs cause "dry mouth" as a side effect?*
**Model Answer:**

```
Many drugs cause xerostomia, or dry mouth.
<|Understanding Medication Side Effects:  The Prevalence of
Dry Mouth in Clinical Trials|>
...can affect comfort and oral function...
<|Understanding Medication Side Effects:  The Prevalence of
Dry Mouth in Clinical Trials|>
...common in antihistamines and antidepressants...
<|Understanding Medication Side Effects:  The Prevalence of
Dry Mouth in Clinical Trials|>
...dosage changes may help manage it...
<|Understanding Medication Side Effects:  The Prevalence of
Dry Mouth in Clinical Trials|>
...
```

**Correctness Score:** 0.0 **Citation Score:** 1.0

The citations are accurate, but the answer never actually addresses why these drugs cause dry mouth, indicating a reasoning gap.

4. "TITLE LURE" ERRORS IN SHORT-FORM QA

In short-form QA tasks, the model sometimes selects citations solely based on title relevance, even when the document content lacks the required evidence. This reflects a superficial attribute mechanism.

**Example:**
**Question**: *How is Boston addressing the digital divide in terms of communications technology from December 2023?* **Model Answer:**

```
Boston is providing subsidized smartphones and free digital literacy workshops.
<|Integrative Technologies in Smart City Infrastructure|>
```

**Ground-truth citation**: *Boston's Infrastructure Development Initiatives: Paving the Way Forward*

While the cited title seems topically aligned, it does not support the factual claim. This failure mode could mislead users who assume correctness based on citation phrasing.

5. CROSS-DOMAIN LOOKALIKES AND "NEAR MISS" CITATIONS

Occasionally, the model cites from a mismatched domain—e.g., a general Wikipedia article instead of a domain-specific source like RepliQA—producing citations that superficially resemble the ground truth but lack factual alignment.

**Example:**
**Question**: *When was the last game of Copenhagen's basketball season played in 2023?* **Model Answer:**

```
The final game was played on 19 May 2023.
<|Copenhagen|>
```

**Ground-truth citation**: *Intramural Sports: A Growing Trend in Copenhagen*

The Wikipedia article Copenhagen is a superficial match but lacks the actual fact. The model retrieves the wrong domain even while giving the correct answer—possibly from memory.

# H   ACTIVE INDEXING

Here we show the details of active indexing.

## H.1   FORWARD

**Step 1: Entity Extractions**   we first extract entities from each document, with the prompt:

> **Entities Extraction**
>
> You will be given a document. Your task is to extract important entities mentioned in the text.
> Entities include names of people, organizations, locations, dates, and other identifiable items. Use the
> categories below as a guide:
> • People & Organizations – Person, Organization
> • Locations – Country, City, Facility, Region
> • Temporal – Date, Time
> • Events – Historical or notable events
> • Objects – Products, Works of art, Laws/Policies
> • Concepts – Theories, Fields, Ideologies
> • Quantities – Numbers, Money, Rankings
> • Biological/Chemical – Species, Compounds
> • Other – Named documents, Tasks, Technologies
>
> Document: [document]
> Only return the 20 most important entities based on their relevance to the main topics of the document,
> where each entity is separated by a newline. In your output, only return the important entities themselves,
> and do not return any other information like their categories or types.

**Step 2: Forward Data Augmentation**   Then we utilize an LLM (Qwen-2.5-7B trained by the seed
data generated from GPT-4.1-mini) to generate relevant questions to each entity conditioned on the
document.

> **Forward Augmentation**
>
> You will be given a document, its title, and an entity from the document. Your task is to generate
> detailed questions that explore the relationship between a given entity and the document. Specifically,
> ask how, what, when, where, why, or if the entity is related to the content of the document. In each of
> your questions, you should include the entity and the document title. The questions should require a
> detailed answer.
> For each question you create, provide a detailed, elaborated answer that explains the relationship
> between the entity and the document. The answer should be based on the content of the document and
> should not include any external information.
> Title: [title] Document: [document]
> Entity: [entity]

**Step 3: Clean up and adjust titles**   : When LLMs generate questions, the document identifiers
may omit parts of the original titles. To correct this, we apply heuristics to locate and replace them
with the corresponding titles from the corpus. We then mark the document identifiers in the questions
with special tokens.

We extract up to 10 entities per document, resulting in approximately 1.28B tokens—3.3× the size of
the original corpus.

## H.2   BACKWARD

**Step 1: Retrieval**   We construct both sparse (BM25) and dense indexes (Lin et al., 2021) using
Pyserini[5]. For each document chunk, we retrieve top-200 relevant chunks using sparse and hybrid
retrieval. From these, we select the top 10 chunks from distinct documents, then randomly sample
1–3 to form a diverse chunk cluster of size 2–4.

---

[5]https://github.com/castorini/pyserini

**Step 2: Generation**   We use GPT-4.1-mini and a fine-tuned Qwen-2.5-3B to generate cross-document instruction-response pairs from the retrieved clusters. Qwen-2.5-3B is trained on GPT-4.1-mini outputs to reduce generation costs. The input prompt format is detailed below.

**Step 3: Filtering**   LLM-generated document identifiers may contain noisy patterns (e.g., "document: xx", "title: xx", or generic placeholders like "document 1"). We detect such cases using heuristics and discard them, which filters out approximately 4.9% of the data. And then we replace the <source> marker with the special tokens.

As a result, we generate around 1.47B tokens (3.8x) for main experiments.

---

**Backward Augmentation**

You will be provided with multiple documents. Your task is to construct a self-contained instruction-answer pair that requires a language model to synthesize information from two or more of these documents in order to generate the correct answer.

The instruction should:
• Be clear, specific, and fully self-contained, so it can be understood without access to or mention of the original documents.
• Prompt the model to integrate, compare, or reason across multiple sources of information.
• Avoid phrases like "in the provided documents", "based on the above", or anything that references the existence of documents.

The answer must:
• Be derived by combining or reconciling information from multiple documents.
• Attribute every factual claim using the format <source>The title of the document</source> to indicate where the information came from.

[documents]

Return the instruction in the first paragraph and the answer in the following paragraphs.

---

### H.3   GENERATOR MODELS CAPACITY ON DATA QUALITY

We investigate whether the effectiveness of Active Indexing depends on the choice of data generator. For backward augmentation, we compare data generated by GPT-4.1-mini versus a 3B model trained by GPT-4.1-mini. Experiments are conducted on RepliQA using LLaMA-3-8B. Citation precision drops from 30.6 to 25.3 when switching from GPT-4.1-mini to the tuned Qwen-2.5-3B generator, despite using more data (34M vs. 40M tokens). Nonetheless, this still significantly outperforms the passive indexing baseline (2.6 citation precision).

## H.4 EXAMPLE ON FORWARD AUGMENTED DATA

---

**Forward Augmentation Example**

Jack Arnold (director)

**Question 1:** How is the television series "Alias Smith and Jones" connected to Jack Arnold's career as described in the document titled <source>Jack Arnold (director)</source>?

**Answer:** According to the document <source>Jack Arnold (director)</source> Jack Arnold directed several episodes of the television series "Alias Smith and Jones" during his television career. This indicates that Arnold was involved in the production of television shows beyond his work in film, including directing episodes of this series that aired in the 1950s. The document highlights Arnold's versatility and his ability to work in different media, including television, alongside his filmmaking career.

**Question 2:** What role did Jack Arnold play in the television series "Alias Smith and Jones," and how does this fit into his overall career as detailed in the document titled <source>Jack Arnold (director)</source>?

**Answer:** The document does not specify the exact role that Jack Arnold played in the television series "Alias Smith and Jones." However, it notes that Arnold directed multiple episodes of this series, which falls under his television work. This placement suggests that Arnold was a director involved in various television productions, contributing to the overall television landscape during the 1950s. His involvement in "Alias Smith and Jones" fits into his broader role as a director who transitioned from film to television, showcasing his versatility and the variety of projects he undertook throughout his career.

**Question 3:** When did Jack Arnold begin directing episodes of "Alias Smith and Jones," and what significance does this timeline have in the context of his career as presented in the document <source>Jack Arnold (director)</source>?

**Answer:** The document does not provide a specific year for when Jack Arnold began directing episodes of "Alias Smith and Jones." However, it places Arnold's work on television, including directing episodes of "Alias Smith and Jones," within his television career, which is noted to have begun in 1955. This timeline situates Arnold's television directing alongside his earlier work on "Science Fiction Theater," "Perry Mason," and other productions, indicating a steady progression into television during the 1950s. This placement highlights the evolution of Arnold's career from film into television as a prominent director.

**Question 4:** Why is "Alias Smith and Jones" important in the context of Jack Arnold's work as discussed in the document titled <source>Jack Arnold (director)</source>?

**Answer:** "Alias Smith and Jones" is important in the context of Jack Arnold's work because it represents one of the television productions he directed, adding to his diverse portfolio of work that includes both film and television. The document emphasizes Arnold's role as a director who worked across genres and mediums, and "Alias Smith and Jones" is part of the evidence of his involvement in television series during the 1950s. This indicates that Arnold was not confined to film but also contributed significantly to television, making him a multifaceted director. ...

---

## H.5 EXAMPLE ON BACKWARD AUGMENTED DATA

---

**Backward Augmentation Example**

Instruction: Explain how the molecular weight and chemical modification of polypropylene influence its compatibility and processing in polymer-clay and polymer-fiber nanocomposites. In your explanation, discuss the role of maleic anhydride grafting and other grafted copolymers as compatibilizers, how molecular weight parameters are characterized, and how these factors affect composite preparation methods and resulting material properties.

Answer: The molecular weight of polypropylene and its chemical modification significantly affect its compatibility and processing behavior in both polymer-clay and polymer-fiber nanocomposites. Maleic anhydride grafted polypropylene (PP-g-MAH), which contains a small amount of grafted maleic anhydride (e.g., 0.29 wt% MAH), is prepared via melt grafting and is commonly used as a compatibilizer to enhance interaction between polypropylene and fillers such as montmorillonite clay. The PP-g-MAH/clay nanocomposites are prepared by melt compounding at temperatures around 190–200 °C, with clay loading around 3 wt% to promote good dispersion and intercalation within the polymer matrix <source>Study on nonisothermal crystallization of maleic anhydride grafted polypropylene/montmorillonite nanocomposite</source>.

Characterization of molecular weight in polymers like polypropylene is essential for understanding how chain length affects viscosity and processing. Molecular weight parameters such as weight-average molecular weight (Mw), number-average molecular weight (Mn), and intrinsic viscosity are typically determined using techniques such as size-exclusion chromatography (SEC) and viscometry in suitable solvents. Studies on poly(1,4-butylene adipate) demonstrate that intrinsic viscosity follows a Mark-Houwink-Sakurada (MHS) type relation with molecular weight, which varies depending on molecular weight distribution breadth. Though this example is for PBA, similar principles apply to polypropylene regarding the relationship between molecular weight and flow properties <source>Relationship of intrinsic viscosity to molecular weight for poly (1, 4-butylene adipate)</source>.

In addition to maleic anhydride grafting, other grafted copolymers such as PP-g-GMA (maleic anhydride replaced by glycidyl methacrylate groups) and SEBS-g-GMA are employed as compatibilizers to improve interfacial adhesion in polypropylene composites with natural fibers like hemp. These grafted compatibilizers are prepared by melt radical grafting and facilitate better bonding between hydrophobic polypropylene and hydrophilic fibers, often used alongside radical initiators in melt mixing processes at temperatures around 170–200 °C <source>Functionalization, compatibilization and properties of polypropylene composites with Hemp fibres</source>.

Overall, molecular weight influences processability by affecting polymer viscosity and chain entanglements, impacting the dispersion of fillers and the mechanical performance of the composite. Chemical modification through grafting functionalities like maleic anhydride or glycidyl methacrylate introduces reactive sites or compatibilizing groups that improve the interfacial adhesion between polypropylene and both inorganic fillers (like montmorillonite clay) and organic fibers (like hemp). These compatibilizers enable the use of melt mixing techniques at elevated temperatures to produce nanocomposites or fiber composites with improved structural and functional properties <source>Study on nonisothermal crystallization of maleic anhydride grafted polypropylene/montmorillonite nanocomposite</source>, <source>Functionalization, compatibilization and properties of polypropylene composites with Hemp fibres</source>.

---

## I  LIMITATIONS AND FUTURE WORK

### I.1  LIMITATION

**Scalability: Model Size, Corpus Coverage, and Augmentation Budget**  Our setup offers a computationally manageable framework for academic research, but it operates at a much smaller scale than production-grade pretraining, which often involves trillions of tokens across vastly more diverse domains than those included in our study. We observe a large performance gap between 3B and 7B models, yet it remains uncertain how these trends evolve at larger scales (e.g.,100B). It is an open question whether Active Indexing continues to yield gains, plateaus, or even regress as model capacity increases. Our data augmentation budget is limited in the main experiments. However, our

scaling analysis shows consistent improvement even at 16× the original data size, suggesting that the method could further benefit from larger augmentation.

**Synthetic Data Quality**  LLM-generated QA pairs may introduce hallucinations into the augmented data, which can lead to subtle degradation in model behavior. While effective in aggregate, this approach could benefit from future work on hallucination detection, filtering, or confidence-aware generation strategies.

**Evaluation Limitations**  Our evaluation of long-form citation relies on NLI models to judge claim support, introducing dependence on their accuracy and coverage. Although this provides a scalable proxy, it adds noise to the measurement and may miss nuanced cases. Incorporating human evaluation would strengthen the reliability of results, particularly for ambiguous or multi-hop claims.

## I.2  FUTURE WORK

**Multilingual and Domain-Specific Attribution**  Our experiments are limited to English and general-domain corpora. Extending Active Indexing to multilingual settings and high-stakes domains—such as law, medicine, or finance—poses unique challenges. These domains often require precise terminology, complex reasoning, and domain-specific citation standards. Future work could develop tailored QA generation methods and identifier formats for these settings, and perform in-depth evaluations of citation fidelity and safety in domain-critical applications.

**Enhancing the existing bindings between facts and their identifiers**  While our methods focus on building new bindings between facts and document identifiers, existing pre-trained models may already encode implicit associations between facts and surface features such as titles, as we observed in the Wikipedia corpus. Beyond titles, there may be other weak or "loose" associations already present in the model, which could potentially be strengthened through better prompting strategies or further post-training methods such as reinforcement learning.

**Scaling Laws and Saturation Points**  We observe consistent improvements with increased augmentation and model size up to 7B, but it remains unclear when and whether gains saturate. With more computational resources, future work can extend scaling curves to larger models (e.g., 14B, 32B, 70B+) and higher augmentation regimes (e.g., 32×, 64×). This would help identify optimal compute-utility tradeoffs and determine whether benefits of Active Indexing persist at frontier scale.

**Complementarity with Retrieval-Augmented Generation (RAG)**  Internal citation and external retrieval are complementary: the former leverages memorized knowledge, while the latter provides up-to-date or unseen information. A promising direction is confidence-aware hybrid systems—where the model cites from internal memory when confident, but falls back to retrieval when uncertain. Exploring how Active Indexing can be integrated into such hybrid systems may yield the best of both worlds: low-latency and high-coverage citation.

**Privacy-Preserving Attribution**  Enabling internal citation increases the model's tendency to surface memorized content, which may include sensitive or proprietary information. Investigating whether attribution behavior exacerbates privacy risks is an important open question. Future work could explore mitigation strategies, such as differential privacy, selective redaction of identifiers, or training-time filtering, to balance attribution fidelity with privacy preservation.

**Human-Centered Evaluation and Interpretability**  While our current evaluation pipeline is largely automatic, the real-world utility of citations depends on user trust and interpretability. Future work could conduct human studies to assess how internal citations affect perceived credibility, transparency, and user trust—particularly in comparison to RAG or non-citing models. Incorporating explanations of why a citation was chosen (e.g., via rationales) could also improve interpretability and debuggability.

## J   USE OF LLMS

We used LLMs to assist with writing. Specifically, we employed GPT-5 thinking, GPT-5 and GPT-4o to rephrase paragraphs for grammatical correctness and improved flow. We also used them to shorten text, making descriptions more concise and easier to read. All LLM-generated text was reviewed, edited, and approved by the human authors.

