# OpenReview forum: "Cite Pretrain: Retrieval-Free Knowledge Attribution for Large Language Models"
_ICLR.cc/2026/Conference — ICLR 2026 Poster_

### Official Review · Reviewer_icLP · 2025-10-30

**Soundness:** 2
**Presentation:** 2
**Contribution:** 2
**Rating:** 4
**Confidence:** 3

**Summary:**

The paper studies whether large language models can produce reliable, verifiable citations to the documents they were (continually) pretrained on, without using test-time retrieval. It introduces a two-stage framework, including continual pretraining with “Active Indexing” plus citation-style instruction tuning and a new benchmark, CitePretrainBench, to evaluate both short- and long-form citation over mixed sources such as Wikipedia, Common Crawl, arXiv, and synthetic unseen documents.

**Strengths:**

1. The problem is well-motivated and distinct from standard RAG: it targets internal, training-time attribution rather than test-time retrieval.
2. The proposed Active Indexing pipeline yields clear, reported gains over passive / repetition baselines on multiple QA-style tasks.

**Weaknesses:**

1. A first concern is cost and practicality: the gains come from heavy, LLM-generated active augmentation, including intra-document QA, cross-document synthesis, bidirectional objectives (scaling to 2.75B extra tokens), which means the approach improves because we invest substantial additional training signal, not necessarily because it is an inherently lightweight replacement for retrieval. A deployment-oriented reader will wonder how often such augmentation must be repeated when the corpus changes.
2. The evaluation benchmark is authored by the paper itself: although it is diverse, it is still tailored to documents with clean, human-readable titles and to a closed identifier space; it remains unclear how well the method transfers to messier enterprise or web-scale settings with noisy IDs, partial documents, or no stable titles.
3. Many of the reported improvements start from very low baselines (e.g., citation precision ≈2–5% for passive or instruction-only models), so relative lifts of “up to 30 points” overstate the absolute reliability; even the best models still lag behind their own answer correctness, and long-form settings (ELI5, ASQA) remain far from robust attribution.
4. Comparison with stronger external-citation / RAG baselines is informative but not exhaustive: the paper shows internal and external are complementary and that a hybrid is best, yet the hybrid still leaves a noticeable gap to the “oracle” combination, suggesting that the proposed training does not fully solve conflict resolution between memorized and retrieved evidence.

**Questions:**

See weaknesses

---

> ### Author Response · Authors · 2025-11-23
> **Author Responses to Reviewer icLP (1/N)**
>
> We appreciate the reviewer’s insightful and helpful feedback.
>
> > W1: Cost and practicability; Not a lightweight replacement for retrieval; How often such augmentation must be repeated
>
> Our “lightweight” claim is about **inference**, not **training**. The active indexing phase is a *one-time* continued pre-training cost on a fixed corpus, whereas RAG pays latency and infra costs on *every query* as mentioned at the end of §5.1.
>
> While our method introduces additional training costs, modern LLM training is increasingly **data-limited** rather than **compute-limited**: the pool of high-quality existing human data is fixed and limited, simply adding compute yields diminishing returns [1][2]. So a popular trend now is to study how to generate more data to enhance model capabilities (e.g, trillion of synthetic tokens in math domains [3]). We believe in this data-constrained regime, our method which trades extra compute for stronger citation ability is well aligned with current industry practice around scaling to ever larger datasets.
>
> Our approaches only need to be run once for a stable corpus. For new fast-changing knowledge, our approach can be combined with external citation: use internal citation for the stable backbone, and external citation for the small, rapidly evolving slice of the knowledge base.
>
> [1] Villalobos et al., Position: will we run out of data? limits of LLM scaling based on human-generated data. ICML ’24 \
> [2] Muennighoff et al., Scaling data-constrained language models. NeurIPS ‘23 \
> [3] Qin et al., Scaling Laws of Synthetic Data for Language Models. COLM ‘25
>
> **Action:** We added a discussion clarifying the current data-constrained regime and how our method fits into and contributes to it ( §5.1). We also included practical deployment guidance for readers.
>
> ---
>
> > W2: transfer to messier enterprise or web-scale settings with noisy ID, partial documents or no stable titles.
>
> The concern that our benchmark depends on “clean titles” is incorrect. Our evaluation already includes **noisy web sources such as Common Crawl, where documents lack reliable IDs or human-readable titles**. For these cases, our method automatically assigns natural-language titles. Because the natural language space is unbounded, we can always generate unique, non-colliding identifiers, as detailed in Appendix A.
>
> **Action:** We further highlight how we deal with the noisy sources like Common Crawl in §3.2.
>
> ---
>
> > W3: up to 30 points overstate the absolute reliability; still lag behind their own answer correctness
>
> The core premise of this concern, that citation “lags behind” correctness and our gains are overstated, is factually incorrect.
>
> 1. **Citation does not lag correctness.**
> On long-form QA, citation is higher: ASQA (30.9 vs. 27.6), ELI5 (29.3 vs. 17.6). On RepliQA, the gap is small (24.4 vs. 31.9). The only outlier is SciQAG, and that gap is a metric artifact: correctness uses a 1–5 partial-credit scale, while citation is strict 0/1. These metrics are not comparable.
> 2. **We did not overstate gains.**
> Our abstract clearly reports improvements **relative to the Passive Index baseline**, and all absolute numbers are shown in the tables transparently.
> 3. **Low absolute scores reflect model limits, not a method issue.**
> A model with modest QA correctness will naturally have modest citation precision, and this is a model capacity limit. Both metrics rise consistently from 3B → 7B → 14B, confirming one bottleneck is model capacity.
> 4. **Our method lifts this ceiling.**
> Scaling Active Indexing data without increasing model size pushes the 7B model to 46.7% citation / 44.6% correctness on repliqa  (§5.3), with gains still unsaturated even at 16× data..  This shows the method is effective and has substantial room to scale.
>
> **Action:**  We clarify that citation precision tracks answer correctness (§5.2), showing the gap reflects model limits, not the method. We also add results from Qwen-2.5-14B, Llama-3.1-8B, and Llama-3.2-3B, confirming both metrics rise monotonically with model capacity (Appendix B)
>
> ---
>
> >  W4: the hybrid still leaves a gap to the oracle; not fully knowledge conflicts.
>
> While further resolving knowledge conflicts is desirable, it is not the focus of our paper. Our goal is to study **internal citation** and show that it combines effectively with external citation, and our results already demonstrate this **complementarity**, even without fully solving conflict resolution.
>
> Pushing the hybrid score higher is always desirable, but fully resolving knowledge conflicts remains an open challenge even for QA correctness [1], not just citation, and is orthogonal to our scope.
>
> [1] Xie et al., "Adaptive chameleon or stubborn sloth: Revealing the behavior of large language models in knowledge conflicts.", ICLR 2024
>
> **Action:** We add a note to §5.3 clarifying that the remaining hybrid–oracle gap reflects the broader open problem of knowledge conflict resolution, which we leave to future work.

---

### Official Review · Reviewer_3iUa · 2025-10-31

**Soundness:** 3
**Presentation:** 3
**Contribution:** 2
**Rating:** 4
**Confidence:** 4

**Summary:**

This paper investigates how to make Large Language Models (LLMs) reliably cite the specific documents from their pretraining data without needing an external retrieval system (e.g. RAG) at inference time. The authors aim to solve the problem of citation hallucination and create a retrieval-free alternative to RAG. The core contribution is a training strategy called **Active Indexing**. The authors contrast this with what's called **Passive Indexing** that has been studied in prior work, where a document identifier (like its title) is simply appended to the end of the text during pretraining. The authors constructed a training corpus by collecting common factual sources such as wikipedia,  then created the CitePretrainBench for evaluation by using four existing question-answering datasets (ASQA, ELI5, SciQAG, and RepliQA) whose answers are grounded in that specific corpus. Their results find supporting evidence of active indexing being a superior method than passive indexing (subject to token replay).

**Strengths:**

The paper systematically conduct empirical analysis on knowledge attribution methods that moves beyond proposing a new method. By directly contrasting simple "Passive Indexing" with their more sophisticated "Active Indexing," the work provides the community with a clear understanding of how different training strategies for attribution directly impact a model's citation accuracy. Furthermore, their scaling experiments offer crucial insights, demonstrating that this citation capability consistently improves with the amount of augmented data, which helps quantify the trade-offs and charts a path for future work in this domain.

**Weaknesses:**

**Novelty Concern**: The proposed training pipeline still largely mirrors Khalifa et al. (2024): (i) continue pretraining on a corpus where each document is tagged with a unique identifier, and (ii) instruction-tune the model to answer questions and emit supporting identifiers, i.e., to ground answers in specific training documents. The paper positions “Active Indexing” as the main advance beyond this baseline. However, much of Backward Active Indexing looks like a scaled-up version of Khalifa’s fact→source supervision: the model is trained on synthetic instructions whose answers integrate facts from multiple documents, and each generated factual statement is annotated with the supporting set of document titles.

Conceptually, this extends Khalifa’s single-document attribution setup to multi-document synthesis and per-claim citation sets, which is present in prior work. See Patel et al. (Towards Improved Multi-Source Attribution for Long-Form Answer)

**Scope and stability of CitePretrainBench**: CitePretrainBench is described as a benchmark that “mixes real-world corpora (Wikipedia, Common Crawl, arXiv) with novel, unseen documents” and evaluates both short-form (single-fact) and long-form (multi-fact, multi-source) citation. The “novel, unseen” part is primarily RepliQA, which the paper characterizes as short-form QA over “fictional, synthetic documents created post-training cutoff (Monteiro et al., 2024).”

In other words, the “new knowledge” is largely imported from an existing dataset that was written after the model’s cutoff, not newly created here. This raises two issues:
- The benchmark’s claim to test the model’s ability to “learn and cite new knowledge” depends on RepliQA still being out-of-distribution for the underlying models. That claim will become weaker as newer base models inevitably ingest RepliQA-like synthetic corpora (or RepliQA itself) during pretraining, which is outside the authors’ control.

- Aside from that post-cutoff synthetic set, the rest of CitePretrainBench reuses existing QA benchmarks (ASQA, ELI5, SciQAG, RepliQA) and established correctness metrics such as Exact Match Recall, Claim Recall, and FreshEval-style LLM grading, plus citation precision/recall that check whether cited titles actually entail each generated claim.

The unification work — consolidating Wikipedia / Common Crawl / arXiv / synthetic docs under a single deduplicated title space, constraining decoding so the model can only cite that space, and evaluating multi-document, per-claim attribution in long-form answers — is valuable infrastructure. That said, the benchmark is still largely built out of pre-existing datasets and inherited metrics, and its “novel knowledge” component is fragile over time.

**Memorization vs. Generalization Claim**: In Section 5.3, you proposed to use FullDoc -> Partial Doc -> GoldQA -> ModelQA as tasks that shift from testing memorization to generalization. This framing seems to assume that “memorization” = being able to associate a document ID with its own content, and “generalization” = being able to recover the same ID when you only see downstream QA-style text. But I’m not convinced that this setup is actually isolating generalization, as opposed to just increasing task difficulty under progressively less evidence.

**Questions:**

1. Your experiments only included Qwen3B and Qwen7B. Does other model families and scales exhibit similar scaling behavior when trained with active indexing? For example, the paragraph "Model Size Matters." in Section 5.2 needs much more than two models to make claims about patterns on model size.

2.  Section 5.1 suggests that when combined together, Active Indexing dataset is much larger than Passive Indexing baseline. Can you reveal how big is the passive indexing dataset in Section 5.1, or make a table to include comparison to how big the instruction tuning baseline dataset is.

3. Building on question 2, the comparison between Passive Indexing and Active Indexing is currently unfair. Despite adding Replay, Passive Indexing training has less fresh tokens than Active Indexing. Do you observe Active Indexing having consistent advantage over passive indexing when the dataset is truncated into the same size as Passive Indexing?

4. Section 5.4 poses an interesting question about the advantage of internal citation when external citation retrieval quality differs. In Figure 3, the x-axis represents retrieval quality. Can you use a paragraph or a figure to explain how we can quantitatively interpret what's called "retrieval quality" here?

---

> ### Author Response · Authors · 2025-11-23
> **Author Responses to Reviewer 3iUa (1/N)**
>
> We truly thank the reviewer for their thoughtful and constructive feedback and the time they invested in the review.
>
> > W1.1: Novelty Concern; Training Pipelines mirror Khalifa; Backward Active Index looks like a scaled-up version of Khalifa
>
> We respectfully clarify that while our pipeline stages (continual pre-training + SFT) are standard, the supervision signal and learning objective are fundamentally different from Khalifa et al. (2024).
>
> 1. **We identify and formalize the core failure mode of Khalifa et al. (2024): their method learns `verbatim-spans → ID`, not true `fact → ID.`**
> Khalifa assumes internal citation fails because models lack explicit doc IDs, so every document is tagged with an ID. On realistic settings like CitePretrainBench, this `verbatim-spans → ID` mapping collapses. Internal citation is not a memorization task: the same fact appears in many paraphrased, heterogeneous contexts, so surface-form matching is insufficient. What we truly need is `fact → ID`, not `verbatim-spans → ID`.
>
> 2. **Backward Active Indexing is not a scaled-up Khalifa; it fundamentally changes the learning objective.**
>    Instead of learning `verbatim-spans → ID`, Backward Active Indexing trains models to map semantic facts to IDs across paraphrases and diverse contexts, with three design goals:
> - **Break surface-form shortcuts:** multiple paraphrases map to the same ID, enforcing semantic invariance.
> - **Document-agnostic applicability:** Active Index provides a single procedure that handles web pages, Wikipedia, papers, and synthetic docs without domain-specific rules.
> - **Scale cleanly**: The pipeline produces training data in a unified, easily scalable way, enabling us to scale the synthetic data (bringing numerous fresh new tokens) without hand-crafted heuristics or bespoke preprocessing.
>
>
> To sum up, this isn’t a simple scaled-up version of Khalifa. Instead, we answer four key questions:
>
> - **Do we need scale?**
>   Yes—otherwise models collapse to verbatim memorization.
>
> - **What should we scale?**
>   Fact-level mappings between non-verbatim statements and doc-IDs.
>
> - **How do we scale efficiently?**
>   By actively teaching the model to use doc-IDs rather than hoping it learns them from passive exposure.
>
> - **How do we scale without saturating quickly?**
>   By synthesizing across related documents, yielding **combinatorially diverse**, high-value fresh tokens, which is something rephrase-only or verbatim methods cannot achieve.
>
> **Action:** We incorporate this discussion into the introduction.
>
>
> ---
>
> > W1.2: Prior work in Multi-source attributions.
>
> Patel et al. perform multi-source attribution in an **open-book** setting: the model cites documents explicitly provided in the context. Our work concerns **internal, closed-book citation**—requiring the model to cite documents purely from parametric knowledge. This is a different problem setting, different supervision signal, and different challenge.
>
> **Action:** We add a comparison to Patel et al. to highlight the contrast between external (open-book) and internal (closed-book) citation in Appendix C.
>
> ---
>
> > W2: Scope and Stability of Cite-Pretrain; RepliQA being included into newer model pre-training; No new data/metric proposed.
>
> **RepliQA Contamination:**  We agree there is a risk that future models may ingest RepliQA, but **its authors explicitly request that it be excluded from pre-training**. Benchmark validity routinely depends on respecting such exclusion lists. Contamination cases (e.g., Qwen-2.5-Math might be trained on Math-500 test set [1]) are treated as **violations of protocol, not evidence that the benchmark is flawed**. If future models ignore this and train on RepliQA, that reflects a lapse in their data curation, not a weakness of the benchmark. We shouldn't discard a rigorous evaluation tool today based on assumed future non-compliance.
>
> **Value of Unification:** We respectfully argue that a benchmark’s value lies in its **diagnostic power**, not the novelty of its raw text. We carefully selected representative datasets and corpus to cover critical axes—short-form vs. long-form, new vs. old knowledge, and doc quality (clean like wiki  vs. noisy like common crawl). By unifying these diverse corpora under a single, deduplicated document-ID space, CitePretrainBench provides the first controlled environment to:
> - Expose Generalization Failures: We reveal that "verbatim spans-to-ID" methods fail in realistic settings, **a critical finding that synthetic-only benchmarks from Khalifa et al. missed.**
> - Compare Paradigms:  This infrastructure enables the first rigorous, side-by-side comparison between Internal Citation and External Citation. Without this unified testbed, it would be impossible to demonstrate our finding that these approaches are complementary.
>
> [1] Wu et al., Reasoning or Memorization? Unreliable Results of Reinforcement Learning Due to Data Contamination. Arxiv, 2025
>
> **Action:** We add above dicussions into §3.2.

---

> ### Author Response · Authors · 2025-11-23
> **Author Responses to Reviewer 3iUa (2/N)**
>
> > W3: Memorization vs. Generalization Claim; Not Isolating Generalization
>
> We thank the reviewer for raising this important question about whether our setup truly isolates generalization rather than merely increasing task difficulty. We address the concern at two levels: (1) clarifying what we mean by generalization, and (2) presenting a new ablation that measures generalization by testing whether the model relies on semantic shortcuts to predict ID or actually learns to use the doc-ID.
>
> **Clarifying our Definition of Generalization.** In our setting, *generalization* refers to the ability to correctly use document IDs under diverse downstream constraints, beyond memorizing verbatim pre-training docs. This requires overcoming two barriers:
>
> - **Evidence Scarcity** (Partial Doc, QA): the model must identify the document from sparse or incomplete input.
> - **Distribution Shift** (QA): the model must identify the correct ID even when the information is paraphrased or presented in a downstream QA format.
>
> Under this definition, the progression **FullDoc → PartialDoc → QA** is not only less evidence, but a controlled way to probe both scarcity and shift. PartialDoc specifically isolates scarcity, while QA measures the additional loss attributable to shift.
>
>
>
> **Isolating Generalization:** We interpret the reviewer’s concern about isolating generalization as asking whether models succeed simply because the statement (question+answer) is semantically similar to the title, rather than because they learned a true `fact→ID` link. To directly test this, we conducted a new ablation using the RepliQA-only setup.
>
> For each statement (query + answer) in RepliQA, we:
>
> 1. Computed the semantic similarity (sentence embeddings) between the statement and all document titles.
> 2. Ranked the true title among them:
>    - **High Rank (e.g., Top-1; high distinctiveness)**:  The target title is more similar to the statement than almost all other titles → easy to shortcut.
>    - **Low Rank**:  Many other titles are more similar → hard to shortcut; requires genuine fact → ID association.
>
> We then stratified examples into four equal-sized bins by the target title rank: *Easy, Medium, Hard, Very Hard*.
> Below we show citation precision and the average target-title rank within each bin:
>
> | Metric    | Easy  | Medium | Hard  | Very Hard | Total |
> |-----------|-------|--------|-------|-----------|-------|
> | C-Pre     | 55.9  | 49.6   | 40.1  | 40.0      | 46.7  |
> | Avg. Rank | 2     | 10     | 60    | 761       | 208   |
>
>
>
> We find:
>
> 1. **Semantic shortcutting is difficult on RepliQA.**
>    Over 90% of target titles in RepliQA have at least one other title more similar to the statement than the true title.
>    The average similarity rank of the true title is `208 / 6,822`, meaning titles are *far from semantically unique*.
>
> 2. **Distinctive titles do make citation easier.**
>    Citation precision is the highest in the Easy bin and decreases as titles become less distinctive, indicating that the model *does* use semantic cues when they exist.
>
> 3. **But performance remains non-trivial even when semantic shortcuts fail.**
>    The model still achieves meaningful C-Pre in the Hard / Very Hard bins (40.1, 40), where semantic similarity gives limited signal. These results confirm that the model learns **fact-level associations**, not merely surface-level shortcuts.
>
> **Action:** We add this experiment to §5.3.

---

> ### Author Response · Authors · 2025-11-23
> **Author Responses to Reviewer 3iUa (3/N)**
>
> > q1: more models
>
> We add results on `Llama-3.1-8B`, `Llama-3.2-3B`, and `Qwen-2.5-14B` to Appendix B. All show the same pattern: active indexing consistently outperforms passive indexing, and performance increases monotonically with model size. Below show results on RepliQA and Eli5. Please see Appendix B for full results.
>
> | Dataset | Metric | **L-8B** |       | **L-3B** |       | **Q-14B** |       | **Q-7B** |       | **Q-3B** |       |
> |---------|--------|----------|-------|----------|-------|-----------|-------|----------|-------|----------|-------|
> |         |        | PassIdx  | ActIdx| PassIdx  | ActIdx| PassIdx   | ActIdx| PassIdx  | ActIdx| PassIdx  | ActIdx|
> | Eli5    | Acc    | 16.1     | 17.9  | 11.2     | 15.6  | 18.5      | 19.3  | 14.5     | 17.6  | 11.7     | 15.8  |
> | Eli5    | C-Pr   | 9.3      | 30.7  | 7.1      | 25.9  | 9.3       | 33.2  | 8.9      | 29.3  | 7.1      | 19.7  |
> | RepliQA | Acc    | 25.6     | 35.7  | 23.4     | 26.8  | 28.4      | 37.0  | 24.8     | 31.9  | 22.8     | 24.5  |
> | RepliQA | C-Pr   | 1.9      | 36.2  | 1.8      | 13.9  | 2.0       | 30.8  | 2.4      | 24.4  | 1.8      | 10.5  |
>
>
>
>
> > q2: Section 5.1 suggests that when combined together, Active Indexing dataset is much larger than Passive Indexing baseline. Can you reveal how big is the passive indexing dataset in Section 5.1, or make a table to include comparison to how big the instruction tuning baseline dataset is.
>
> The total active indexing in the main experiments include 2.75B tokens, which is around *7.05 (390M original corpus tokens). We make it more explicit in section 5.1.
>
>
>
> > q3: Building on question 2, the comparison between Passive Indexing and Active Indexing is currently unfair. Despite adding Replay, Passive Indexing training has less fresh tokens than Active Indexing. Do you observe Active Indexing having a consistent advantage over passive indexing when the dataset is truncated into the same size as Passive Indexing?
>
> We addressed the concern about "fresh tokens" in Section 5.3. We compared Active Indexing against **Passive Indexing + Synthetic Continual Pretraining (SCP)**. SCP is a strong augmentation baseline that injects fresh, rephrased tokens by exploiting entity relationships within each document and produces substantially more diverse paraphrases than standard rephrasing methods. Despite this, **Active Indexing consistently outperforms PI+SCP as fresh tokens increase (Fig. 2)**. PI+SCP is less token-efficient because it still never teaches the model to use document identifiers.  **PI+SCP also saturates more quickly.** Active Indexing continues to improve because it produces richer, more diverse fact-level tokens by synthesizing across documents, which yields a combinatorial number of high-value variations that document-local augmentations like SCP cannot achieve.
>
> **Fresh Tokens are Necessary, Not a Confounder**: The reviewer’s fairness concern assumes that we should match the amount of tokens across methods. We see this differently: **the ability to generate and scale the right kind of fresh tokens is exactly our methodological innovation**. Passive Indexing in Kahalifa fails precisely because it lacks this supervision and collapses into verbatim span→ID memorization. Internal citation requires mapping many surface forms of the same fact to the correct ID. Forcing identical synthetic data across methods would remove the very ingredient that distinguishes Active Indexing and would recreate the failure mode we diagnose.
>
>
>
>
>
>
> > q4: retrieval quality explanations
>
> Thanks for raising this point. When we say retrieval quality = 0, we use a purely sparse retriever (BM25) and take its top-5 results. When retrieval quality = 1, we use a purely dense retriever and take its top-5 results. For intermediate values (e.g., 0.2), we create a mixture: 20% of the retrieved documents are sampled from the dense top-5, and the remaining 80% are sampled from the BM25 top-5. We add these detailed explanations in section 5.

---

### Official Review · Reviewer_m27s · 2025-10-31

**Soundness:** 2
**Presentation:** 3
**Contribution:** 1
**Rating:** 4
**Confidence:** 5

**Summary:**

This paper introduces CitePretrain, a method for enabling LLMs to perform reliable internal citations—attributing answers to documents seen during continual pretraining without requiring test-time retrieval. The authors construct CitePretrainBench, a benchmark mixing real-world corpora (Wikipedia, Common Crawl, arXiv) with novel documents for both short-form and long-form citation QA tasks. Their approach uses two-stage training: (1) continual pretraining to index factual knowledge by binding it to document identifiers, and (2) instruction tuning to elicit citation behavior. They propose Active Indexing, which augments training with synthetic data that restates facts in diverse forms and enforces bidirectional training (source→fact and fact→source), significantly outperforming Passive Indexing (simply appending identifiers). Experiments on Qwen-2.5-7B/3B show Active Indexing improves citation precision by up to 30.2%, with performance scaling with augmentation data (up to 16×) and model size. The paper also demonstrates that internal and external citations are complementary, with hybrid approaches performing best across varying retrieval quality.

**Strengths:**

1.  Unlike prior work limited to synthetic biographies, CitePretrainBench introduces the first rigorous benchmark emphasizing real-world complexity across Wikipedia, Common Crawl, arXiv, and novel documents, with both short-form and long-form citation tasks.

2. Active Indexing is conceptually elegant and addresses passive indexing's core limitation: failure to ground non-verbatim content. By generating diverse synthetic QA pairs and enforcing bidirectional training (source→fact, fact→source), it achieves substantial improvements (up to 30.2% citation precision gains). Performance scales favorably—continuing to improve at 16× augmentation without saturation, indicating significant headroom.

3. Thorough ablations reveal that both fact variation and active supervision are essential. The paper demonstrates that internal citations excel under poor retrieval, while external citations perform better with strong retrieval, with hybrid approaches achieving the best overall performance, establishing practical value for real-world deployment.

**Weaknesses:**

(A) The core motivation is weak. The paper leans on the idea that RAG is too expensive or impractical, yet in real deployments RAG is the default pattern to ground models on fast-changing enterprise data. With reranking, caching, and short contexts, teams control tokens and latency, and every major cloud ships managed RAG stacks. If the goal is to argue for internal citation on cost, the comparison needs to be against optimized, production RAG baselines rather than token budget.


(B) The claim that the method “goes beyond memorization” is overstated. Inference is constrained to titles or identifiers, which functions as an ID proxy. The paper itself notes title-level memorization on Wikipedia and infeasibility where stable IDs are weak. Bidirectional training may help usage, but the supervision still steers toward ID cues, which limits granularity.

(C) The paper does attempt to answer why the method works with some ablations and scaling results, but lacks intuitive ablation experiments like ablating title distinctiveness or using near-duplicate titles.

(D) Baselines and model diversity are thin. Main results are on Qwen-2.5 3B and 7B, plus a prompt-only GPT-4.1 check. There is no head-to-head against LongCite-8B or LongCite-9B or other long-context citation-tuned systems, despite long-form citation being a stated focus.

(E) The hybrid story is sensible engineering, but does not address the most important failure case. The paper shows Internal-only, External-only, and Hybrid setups and average complementarity across retrieval quality, but it does not isolate the conflict slice where internal is wrong and external is correct. That is the production reality that needs to be quantified, with an arbitration policy that shows when the system will switch, abstain, or defer.

**Questions:**

Q1 Why are GPT-4.1 scores not bolded where it has the best performance in Table 1, given the caption says “best results are bolded”?

 Q2 What were the reasons behind choosing only Qwen models for evaluation?

 Q3 Why are models like LongCite-8B and LongCite-9B, which are fine-tuned for long-context citation, not included as baselines when long-form citation is a major theme?

 Q4 What happens when internal citations are wrong but retrieved external context is correct and contains the right citation?

---

> ### Author Response · Authors · 2025-11-23
> **Author Responses to Reviewer m27s (1/N)**
>
> We sincerely thank the reviewer for their constructive and insightful feedback.
>
> Regarding:
>
> > W-A: Weak Motivation on RAG being too expensive/impractical
>
> 1. Our goal is not to replace RAG but to study a complementary capability. Internal citation targets cases where models already answer correctly in closed-book mode and we want low-latency, infrastructure-free attribution. Additionally, **standalone modes are still widely used in practice (e.g., ChatGPT with or without web search mode)**, and internal citation can also work alongside RAG to mitigate retrieval errors, as shown in §5.3.
>
> 2. From a scientific perspective, internal knowledge attribution is interesting in its own right. **LLMs can internalize vast factual knowledge, but why can't they internalize sources and still hallucinate citations?** Understanding why models can store facts but not their sources is a fundamental question. Moreover, as you mentioned, RAG pipelines require many engineering components ( reranking, caching, and short contexts, teams control tokens), and these need to be optimized by case. Our method provides a **unified, end-to-end** alternative that can be applied consistently across document types. **Deep learning has historically benefited from replacing handcrafted pipelines with end-to-end models**, as seen in the shift from modular dialogue systems to unified chatbots and from task-specific BERT architectures to general-purpose decoders. We view internal citation as aligned with this long-term direction.
>
> **Action:** we further highlight these points in the introduction.
>
> ---
>
> > W-B: Go beyond memorization is overstated; supervision steer towards ID cues \
> > W-C: Lack of ablation about title distinctiveness and near-duplicate titles
>
> We interpret (B) and (C) as pointing to the same concern: the model might simply memorize titles during training and rely on “ID cues” (semantic similarity between a fact and its title) at test time, rather than truly learning to use doc-ids. Below, we clarify what we mean by "beyond memorization" and then address your request for an ablation on title distinctiveness / near-duplicate titles.
>
> **Clarifying Memorization and Generalization**. We define `memorization` as learning the link between a *specific verbatim pre-training text* and a Doc-ID.  `Generalization` goes beyond this by linking the ID to the **underlying fact itself**, enabling the model to cite the correct source even when the information is phrased differently in downstream tasks.
> To avoid confusion, we revised the “go beyond memorization” framing to be more explicit in the introduction: “To address this, we propose Active Indexing—a method that doesn’t just memorize verbatim text-to-ID links, but teaches the model to recognize and cite the right document even when the underlying fact is variously expressed.”
>
> **Addressing the “Shortcut ID Cues” Hypothesis.** To directly test whether the model relies on ID cues (semantic shortcuts), we conducted a new ablation on the RepliQA-only setup, analyzing `Title Distinctiveness` as you proposed.
>
> For each statement (query + answer) in RepliQA, we:
>
> 1. Computed the semantic similarity (sentence embeddings) between the statement and all document titles.
> 2. Ranked the true title among them:
>    - **High Rank (e.g., Top-1; high distinctiveness)**:  The target title is more similar to the statement than almost all other titles → easy to shortcut.
>    - **Low Rank**:  Many other titles are more similar → hard to shortcut; requires genuine fact → ID association.
>
> We then stratified examples into four equal-sized bins by the target title rank: *Easy, Medium, Hard, Very Hard*.
> Below we show citation precision and the average target-title rank within each bin:
>
> | Metric    | Easy  | Medium | Hard  | Very Hard | Total |
> |-----------|-------|--------|-------|-----------|-------|
> | C-Pre     | 55.9  | 49.6   | 40.1  | 40.0      | 46.7  |
> | Avg. Rank | 2     | 10     | 60    | 761       | 208   |
>
>
>
> We find:
>
> 1. **Titles already exhibit near-duplicate conditions.**
>    Over 90% of target titles in RepliQA have at least one other title more similar to the statement than the true title.
>    The average similarity rank of the true title is `208 / 6,822`, meaning titles are *far from semantically unique*.
>
> 2. **Distinctive titles do make citation easier.**
>    Citation precision is the highest in the Easy bin and decreases as titles become less distinctive, indicating that the model *does* use semantic cues when they exist.
>
> 3. **But performance remains non-trivial even when semantic shortcuts fail.**
>    The model still achieves meaningful C-Pre in the Hard / Very Hard bins (40.1, 40), where semantic similarity gives limited signal. These results confirm that the model learns **fact-level associations**, not merely surface-level shortcuts.
>
> **Action:** We add this experiment to §5.3.

---

> ### Author Response · Authors · 2025-11-23
> **Author Responses to Reviewer m27s (2/N)**
>
> > W-D + Q2 + Q-3: Baselines and model diversity are thin. No comparison to LongCite-8B and LongCite-9B; main theme is long-form citations
>
> We add results on `Llama-3.1-8B`, `Llama-3.2-3B`, and `Qwen-2.5-14B`. All show the same pattern: active indexing consistently outperforms passive indexing, and performance increases monotonically with model size. Below show results on RepliQA and Eli5. Please see Appendix B for full results.
>
> | Dataset | Metric | **L-8B** |       | **L-3B** |       | **Q-14B** |       | **Q-7B** |       | **Q-3B** |       |
> |---------|--------|----------|-------|----------|-------|-----------|-------|----------|-------|----------|-------|
> |         |        | PassIdx  | ActIdx| PassIdx  | ActIdx| PassIdx   | ActIdx| PassIdx  | ActIdx| PassIdx  | ActIdx|
> | Eli5    | Acc    | 16.1     | 17.9  | 11.2     | 15.6  | 18.5      | 19.3  | 14.5     | 17.6  | 11.7     | 15.8  |
> | Eli5    | C-Pr   | 9.3      | 30.7  | 7.1      | 25.9  | 9.3       | 33.2  | 8.9      | 29.3  | 7.1      | 19.7  |
> | RepliQA | Acc    | 25.6     | 35.7  | 23.4     | 26.8  | 28.4      | 37.0  | 24.8     | 31.9  | 22.8     | 24.5  |
> | RepliQA | C-Pr   | 1.9      | 36.2  | 1.8      | 13.9  | 2.0       | 30.8  | 2.4      | 24.4  | 1.8      | 10.5  |
>
>
> Regarding LongCite-8B/9B: these systems are designed for **external citation**, they rely on explicit long-context input and optimize citation given retrieved evidence. Our focus is the complementary question of **internal citation**: can a model cite accurately without context? Because LongCite is not trained for closed-book settings and requires context, a head-to-head comparison is not meaningful.
>
> **Action:** We added Qwen-2.5-14B and Llama-3.2-3B, Llama-3.1-8B to strengthen results and observe similar findings (Appendix B). We further emphasize the difference between external citation methods like LongCite and our internal citation methods in Appendix C.

---

> ### Author Response · Authors · 2025-11-23
> **Author Responses to Reviewer m27s (3/N)**
>
> > **W-E:** Does not isolate the conflict slice where internal is wrong and external is correct.
>
> While `external-correct / internal-wrong` conflicts are a critical production scenario, the reverse (`external wrong / internal correct`) is also common [1]: retrieval can fail, be only partially relevant, or distract the model even when the right document is present [2]. Thus, **both** conflict directions are important production realities that any arbitration policy must handle.
>
> To address this, we now explicitly isolate and analyze conflict slices on RepliQA by partitioning examples into:
> - **E✓/I✗**: External correct, Internal wrong
> - **I✓/E✗**: Internal correct, External wrong
> - **No-conf**: both correct or both wrong  (No conflict)
>
> We then report accuracy for Internal-only, External-only, and hybrid methods under high-quality retrieval (RQ = 1.0) and noisy retrieval (RQ = 0.2).
>
> **Retrieval Quality = 1.0**
> | Method         | E✓/I✗ (38.6%) | I✓/E✗ ( 8.3%) | No-conf ( 53.1%) | Total |
> |----------------|:-----:|:-----:|:-------:|:-----:|
> | Hybrid-Joint   |  75.6 |  51.8 |   48.4  |  59.2 |
> | Hybrid-Fallback|  99.5 |   6.0 |   44.4  |  62.5 |
> | Hybrid-Oracle  | 100.0 | 100.0 |   44.4  |  70.0 |
> | External-only  | 100.0 |   0.0 |   44.4  |  62.2 |
> | Internal-only  |  0.0  | 100.0 |   44.4  |  31.9 |
>
> When retrieval is strong, the dominant conflict is the reviewer’s case (**External correct / Internal Wrong**, 38.6% vs 8.3%). In this slice, Hybrid-Fallback effectively behaves as a “switch-to-external” policy (99.5, matching External-only), while Hybrid-Joint is more conservative but still substantially improves over Internal-only.
>
> **Retrieval Quality = 0.2**
>
> | Method         | E✓/I✗ (9.9%) | I✓/E✗ (22.0%) | No-conf (68.1%) | Total |
> |----------------|:-----:|:-----:|:-------:|:-----:|
> | Hybrid-Joint   |  66.7 |  58.2 |   24.5  |  36.1 |
> | Hybrid-Fallback|  88.9 |  41.8 |   21.9  |  32.9 |
> | Hybrid-Oracle  | 100.0 | 100.0 |   14.5  |  41.8 |
> | External-only  | 100.0 |   0.0 |   14.5  |  19.8 |
> | Internal-only  |  0.0  | 100.0 |   14.5  |  31.9 |
>
> When retrieval is weaker, the dominant conflict flips to **Internal Correct/External Wrong** (22.0% vs 9.9%), and both hybrid methods **shift toward internal knowledge**, substantially outperforming External-only in this slice.
>
> Interestingly, even in the “non-conflicting” slice, Hybrid-Joint (24.5) can outperform both Internal-only and External-only (14.5) by stitching together partial clues from each source, showing a second layer of **complementarity beyond simply picking “internal vs external.”**
>
> Overall, this analysis makes the arbitration behavior explicit: when retrieval is reliable, hybrids act like switch-to-external policies; when retrieval is noisy, they lean back on internal knowledge while still combining both signals.
>
> [1] Wang et al., “Astute RAG: Overcoming Imperfect Retrieval Augmentation and Knowledge Conflicts for Large Language Models”. ACL 2025
> [2] Yoran et al., “Making Retrieval-Augmented Language Models Robust to Irrelevant Context”. ICLR 2024
>
> **Action:** We add these results and analysis to Appendix B.3.
>
> ---
>
> > Q1: Why gpt-4.1 is not bolded
>
> Thanks for pointing this out. The comparisons are meant for the methods within the same model.
>
> **Action:** We have added notes to the main table.
>
>  > Q2: Why choose only Qwen-2.5 3B and 7B
>
> Addressed in response to Weakness-D
>
> > Q3: Why not LongForm-8B/9B
>
> Addressed in response to Weakness-D
>
> > Q4: What happens when internal citations wrong when external correct
>
> Addressed in response to Weakness-E

---

### Official Review · Reviewer_grma · 2025-11-01

**Soundness:** 3
**Presentation:** 3
**Contribution:** 2
**Rating:** 8
**Confidence:** 5

**Summary:**

This paper introduces CitePretrain, a framework that teaches large language models to cite the documents they learned from during pretraining without RAG. The goal is to make models more transparent and trustworthy by letting them show where their knowledge originally came from. The work builds on Khalifa et al. 2024, extending it from synthetic to real-world data like Wikipedia, Common Crawl, and arXiv. The main addition is Active Indexing, a continual pretraining step that links each document to an identifier and trains the model in two directions: Forward indexing: generate facts when conditioned on a document ID and Backward indexing: cite the correct document given a fact. Experiments on Qwen-2.5-3B and 7B show that Active Indexing improves citation precision over baselines, while keeping answer accuracy stable.

**Strengths:**

1. **Well-motivated:**  Making LLMs capable of citing their pretraining data is an exciting direction that’s still largely unexplored. Also, this paper extends prior work in a meaningful way. Moving from synthetic experiments to real-world data and introducing Active Indexing feels like a natural next step.

2. **Active Indexing works well:**  The dual training setup (forward + backward) leads to solid citation gains. The authors also provide detailed ablations showing how much each component contributes, which helps clarify *why* the method works.

3. **Thorough evaluation:** The experiments cover a range of QA datasets and both correctness and citation metrics.

4. **deep analysis and insights**: The discussion on scaling, memorization vs. grounding, and the effects of dataset size is nice to see.

**Weaknesses:**

1. **Heavy additional training cost:**  The method needs a full active indexing phase with two objectives and lots of synthetic QA generation. That’s quite resource-heavy, and it might be hard to apply at the scale of large, from scratch, pretraining runs.

2. **Forward indexing might be unnecessary:**  The results show that backward indexing (fact → doc) already gives most of the gains. The extra forward indexing adds complexity and cost for very little improvement, which makes me question whether it’s worth the extra training. This could be explained by that in forward indexing, teaching the model to go from document ID → facts feels artificial. It’s not something the model will ever be asked to do at inference time, so it might encourage memorization rather than true grounding.

4. **Shortcut risk via document titles:**  Using real document titles as identifiers can leak information. The model might just learn to guess the title instead of genuinely retrieving it. For example, if the question mentions “Barack Obama,” it could easily predict `<s> Barack Obama</s>` as the citation without any retrieval.

6. **Limited OOD evaluation**
   The authors do not evaluate the LM out-of-domain after their active indexing. Khalifa et al., 2024 report increase in perplexity over natural text due to training on document IDs. Do you observe the same? Can you  alsoevaluate the model on OOD tasks e.g., instruction following or QA before and after the indexing training?

**Questions:**

- Do you make sure that you're evaluating citation on totally unseen documents data as in Khalifa et al.? Or could the same document be included in both Active indexing and in the test set?
- Have you tried anonymizing or hashing document titles to make sure the model isn’t just predicting them directly?

---

> ### Author Response · Authors · 2025-11-23
> **Author Responses to Reviewer grma (1/N)**
>
> We appreciate the reviewers’ insightful feedback and their positive view of our work; it truly means a lot to us.
>
> Regarding
>
> > W1: Heavy additional training cost
>
> While our method introduces additional training costs, modern LLM training is increasingly **data-limited** rather than **compute-limited**: the pool of high-quality existing human data is fixed and limited, simply adding compute yields diminishing returns [1][2]. So a popular trend now is to study how to generate more data to enhance model capabilities (e.g, trillion of synthetic tokens in math domains [3]). We believe in this data-constrained regime, our method which trades extra compute for stronger citation ability is well aligned with current industry practice around scaling to ever larger datasets.
>
>  Additionally, the synthetic generation step does not require a strong model; for example, our pipeline includes a fine-tuned Qwen-2.5-3B model for data generation.
>
> [1] Villalobos et al., Position: will we run out of data? limits of LLM scaling based on human-generated data. ICML ’24 \
> [2] Muennighoff et al., Scaling data-constrained language models. NeurIPS ‘23 \
> [3] Qin et al., Scaling Laws of Synthetic Data for Language Models. COLM ‘25
>
> **Action:** We added a discussion clarifying the current data-constrained regime and how our method fits into and contributes to it ( §5.1).
>
> ---
>
> > W2: Forward indexing might be unnecessary.
>
> While Backward Indexing provides the bulk of the gains, Forward Indexing consistently adds value. On RepliQA, we add a new experiment where we scale both directions and their combination, and observe that **each method yields steady improvements as data grow, and their effects are additive when used together**. For users who prioritize absolute performance, this reliable gain can justify the additional compute.
>
> Moreover, Forward Indexing is not artificial. It supports answering realistic queries like “Describe the method and results in paper Y” in a closed-book manner, which requires the model to internally condition on document identifiers. In this sense, Forward Indexing enhances the model’s general ability to use IDs as semantic anchors, not just for citation.
>
> **Action:** We added a new experiment and expanded the discussion to clarify that Forward Indexing offers modest but consistent improvements and enables useful closed-book behaviors beyond citation (Appendix B.2).
>
> ---
>
> > W3: Shortcut risk via document titles
>
> To directly test whether the model is using semantic shortcuts, we conducted a new ablation using the RepliQA-only setup.
>
> For each statement (query + answer) in RepliQA, we:
>
> 1. Computed the semantic similarity (sentence embeddings) between the statement and all document titles.
> 2. Ranked the true title among them:
>    - **High Rank (e.g., Top-1; high distinctiveness)**:  The target title is more similar to the statement than almost all other titles → easy to shortcut.
>    - **Low Rank**:  Many other titles are more similar → hard to shortcut; requires genuine fact → ID association.
>
> We then stratified examples into four equal-sized bins by the target title rank: *Easy, Medium, Hard, Very Hard*.
> Below we show citation precision and the average target-title rank within each bin:
>
> | Metric    | Easy  | Medium | Hard  | Very Hard | Total |
> |-----------|-------|--------|-------|-----------|-------|
> | C-Pre     | 55.9  | 49.6   | 40.1  | 40.0      | 46.7  |
> | Avg. Rank | 2     | 10     | 60    | 761       | 208   |
>
> We find:
>
> 1. **Semantic shortcutting is difficult on RepliQA.**
>    Over 90% of target titles in RepliQA have at least one other title more similar to the statement than the true title.
>    The average similarity rank of the true title is `208 / 6,822`, meaning titles are *far from semantically unique*.
>
> 2. **Distinctive titles do make citation easier.**
>    Citation precision is the highest in the Easy bin and decreases as titles become less distinctive, indicating that the model *does* use semantic cues when they exist.
>
> 3. **But performance remains non-trivial even when semantic shortcuts fail.**
>    The model still achieves meaningful C-Pre in the Hard / Very Hard bins (40.1, 40), where semantic similarity gives limited signal. These results confirm that the model learns **fact-level associations**, not merely surface-level shortcuts.

---

> ### Author Response · Authors · 2025-11-23
> **Author Responses to Reviewer grma (2/N)**
>
> > w4: Limited OOD Evaluations
>
> To address your concern, we run the OOD experiments before/after indexing training and do observe the same increase in perplexity reported by Khalifa et al. Specifically, we evaluate Wikipedia/ArXiv perplexity and TriviaQA accuracy after RepliQA-only continual pretraining (CPT) under several indexing variants. We attribute the increase to two factors:
> (a) **domain-specific CPT**, which is known to induce catastrophic forgetting when the CPT corpus is narrower than the original pre-training data; and
> (b) the **use of doc IDs**.
>
> To disentangle these effects, we start from the same backbone and train on RepliQA only, for the same number of tokens, under five variants:
> 1. `raw data (no titles)`,   2. `Passive Index with natural-language title IDs`,   3. `Passive Index with integer IDs`,   4. `Passive Index with repeated natural-language IDs`,   5. `Active Index`.
>
> We then measure Wiki/ArXiv perplexity and TriviaQA accuracy.
> | CPT Method                     | Wiki Perplexity | ArXiv Perplexity | TriviaQA |
> |-------------------------------|-----------------|------------------|----------|
> | Base Model (No CPT)           | 6.71            | 4.61             | 56.0       |
> | Raw Data (No Title)           | 32.15           | 6.65             | 50.8     |
> | Passive Index (Title ID)      | 54.31           | 6.90             | 49.5     |
> | Passive Index (Integer ID)    | 60.15           | 6.88             | 49.2     |
> | Passive Index Repeat (Title)  | 39.78           | 6.80             | 51.2     |
> | Active Index                  | 26.52           | 5.40             | 50.5     |
> We find:
>
> - **All CPT variants raise OOD perplexity** relative to the base model, as expected from domain-specialized CPT. This effect appears even in the *raw data (no titles)* condition, indicating that most of the degradation comes from narrow-domain CPT rather than indexing itself.
>
> - **Natural-language IDs are less harmful than integer IDs.** Passive Index with integer IDs yields the highest Wiki perplexity, while natural-language titles perform better, suggesting that well-formed text identifiers integrate more naturally into language modeling.
>
> - **Repeating titles is not particularly damaging.** Repeating natural-language titles does not hurt perplexity more than appending them once, unlike Khalifa et al.’s observation. We suspect this difference arises because our identifiers are natural text, not opaque tokens.
>
> - **Active Indexing is the least harmful among ID-based methods.** It achieves the lowest perplexity on both Wikipedia and ArXiv across all RepliQA-only CPT variants, indicating that the augmentation scheme improves fluency within this knowledge domain.
>
> - **Perplexity shifts do not correspond to QA degradation.** Despite variation in perplexity across variants, TriviaQA accuracy of different indexing strategies after fine-tuning remains very similar. This suggests that moderate perplexity changes reflect stylistic fluency or domain fit rather than loss of knowledge.
>
> **Action:** We add the above OOD analysis to Appendix B.4
>
> ---
>
> >  q1: Do you make sure that you're evaluating citation on totally unseen documents data as in Khalifa et al.? Or could the same document be included in both Active indexing and in the test set?
>
> We follow Khalifa et al. in exposing all document identifiers during the continual pretraining indexing phase;  without doing so, the model has no signal from which to learn ID prediction. During fine-tuning, we ensure that the citation targets appearing in the training set have no overlap with those used in the test set.
>
>
> > q2: Have you tried anonymizing or hashing document titles to make sure the model isn’t just predicting them directly?
>
> In Appendix F.1, we compare multiple types of document identifiers, including integer IDs. We find that, under the same learning rate, the model can memorize natural-text titles far more reliably than arbitrary integer identifiers. Making integer IDs learnable requires substantially increasing the learning rate, which in turn degrades language modeling quality. For this reason, we use text titles consistently across all experiments. As we show in our response to Weakness 3, even with natural text titles the model does not rely solely on semantic similarity. It learns the underlying fact→ID associations, rather than just matching surface forms.

---

> > ### Comment · Reviewer_grma · 2025-11-27
> >
> > Thank you for the rebuttal. I'm maintaining my already positive assessment of the paper. Good luck!

---

### Author Response · Authors · 2025-11-26
**Summary of Changes Made in Response to Reviews**

We sincerely thank all reviewers for their thoughtful and constructive feedback.

We are glad that reviewers found the problem of enabling internal citation generation both *exciting* and *well-motivated* (``grma``, ``icLP``). We also appreciate the positive remarks on our *elegant design* (``m27s``), the *significant gains* from Active Indexing (``grma``, ``m27s``, ``3iUa``, ``icLP``), the *thorough evaluation* enabled by CitePretrainBench (``grma``, ``m27s``), our *comprehensive ablations and insights* into how internal citation works (``grma``, ``m27s``, ``3iUa``), and the *practical value* of combining internal and external citations (``m27s``).

Following the feedback, we have revised the manuscript accordingly. All changes are marked in blue.

**New Experiments Added**
- A new ablation showing that Active Indexing is not merely predicting titles via superficial semantic cues but genuinely learns to use document IDs (Sec. 5.3)   (``grma``, ``m27s``, ``3iUa``)
- Additional results on Llama-3.1-8B, Llama-3.2-3B, and Qwen-2.5-14B (App. B.1)   (``m27s``, ``3iUa``)
- An experiment analyzing model behavior across different conflict slices when combining internal and external citation (App. B.3)  (``m27s``)
-  Evidence showing the consistent additive effect of Forward Index under scaling (App. B.2)   (``grma``)
- Additional out-of-distribution analysis experiments (App. B.4)  (``grma``)

**Other Improvements**

We have addressed all detailed concerns and suggestions in the individual reviewer responses, clarifying definitions, tightening presentation, and improving overall writing quality. Each comment received an explicit action.

---

We hope these updates meaningfully address all concerns, and we kindly ask reviewers to reconsider their scores. We believe our work takes an important step toward understanding **why LLMs can internalize knowledge yet struggle to internalize its sources, and how to bridge this gap**. We are happy to clarify any remaining questions.

---

### Author Response · Authors · 2025-12-02
**Rebuttal Summary for Area Chair (1/2)**

We truly appreciate AC's efforts in reviewing. We also thank reviewers for their constructive feedback. Below is a summary of how we addressed the major concerns, organized by Method Validity (does it work), Positioning (does it matter), Practicality (is it practical), and additional clarifications of misunderstandings.

## 1. Method validity and robustness

> `grma-w3`, `m27s-w2+w3`, `3iUa-w3` (Memorization vs. generalization)
Reviewers questioned if the model genuinely learns to utilize Doc-IDs (generalization) or merely relies on "semantic shortcuts" (e.g., predicting a doc-ID like "Barack Obama" based on the "Barack Obama" in the query text, rather than truly generalizing).

**Re:**  We added an ablation in Section 5.3 to demonstrate that Active Indexing is not merely predicting titles via superficial semantic cues but genuinely learns the mechanism of mapping facts to doc-IDs.

---

> `m27s-w4`, `3iUa-q1` (More model families and sizes)
> Concern that main experiments rely primarily on Qwen-2.5 3B and 7B.

**Re:**  We added results for Llama-3.1-8B, Llama-3.2-3B, and Qwen-2.5-14B (App. B.1) to show that the findings are consistent.

---

> `grma-w4` (Limited OOD evaluation)
> Concern about how Active Indexing affects OOD tasks.

**Re:**  We added an OOD analysis in App. B.4.

---

## 2. Positioning, motivation, and novelty

> `m27s-w1` (Motivation vs. RAG-dominated deployment)
> Concern that RAG is the default deployment pattern and therefore internal citation may be weakly motivated.

**Re:**  We clarify that Internal Citation **complements**, rather than **replaces** RAG.
1. Deployment Reality: Standalone models (e.g, OpenAI without web-search mode) are still widely deployed for efficiency.
2. Complementary Strengths: Internal citation offsets retrieval noise while external citations (RAG) provide new knowledge (proven in Sec 5.3).
3. Scientific Value: Internal citation studies an important research question -- **LLMs can internalize vast factual knowledge, but why can't they internalize sources and still hallucinate citations?**  Solving this moves us closer to unified, **end-to-end** systems rather than handcrafted RAG pipelines. Such an end-to-end goal aligns with how deep learning is usually advanced.

**Note:** Reviewers `grma` and `icLP` explicitly cite the problem as well motivated.

---

> `3iUa-w1` (Novelty vs. Khalifa et al.)
> Concern that Active Indexing follows a similar continual-pretraining + SFT pipeline as Passive Index and may be just a scaled-up variant.

**Re:**  While both utilize a CPT+SFT pipeline, our **learning objectives and design are fundamentally different**. We identify and solve the core failure modes of passive indexing. We are not simply scaling up, but we answer **do we need to scale, what to scale, how to scale token-efficiently and non-saturatingly**

**Note:** Reviewer `m27s` calls our design "conceptually elegant" and `grma` notes it "extends prior work in a meaningful way". All reviewers acknowledge the significant improvements of Active Index over Passive Index.

---

## 3. Practicality and efficiency

> `grma-w1`, `icLP-w1` (Training cost and practicality)
> Concern that Active Indexing introduces heavy additional training cost due to extra data.

**Re:**  We argue that current LLM deployment is increasingly **data-limited rather than compute-limited**. As high-quality human data saturates, simply adding compute yields diminishing returns. Our approach aligns with the emerging practice of scaling synthetic data to convert compute into quality. Active Indexing is a **one-time training cost** that amortizes over all future queries, in contrast to RAG, which incurs **recurring retrieval and indexing costs at inference time**.

---

> `grma-w2` (Necessity of Forward Indexing)
> Concern that Backward Indexing provides most of the benefit, so Forward Indexing may be unnecessary.

**Re:**  We added experiments in App. B.2 showing that Forward Indexing provides a **consistent additive effect** under scaling.

---

> `m27s-w5` (Hybrid behavior under knowledge conflict)
> Concern that Sec 5.3 does not isolate cases where internal and retrieved knowledge conflict, which is important for practical use of the hybrid.

**Re:**  We added an experiment analyzing model behavior across different conflict slices (App. B.3), providing a fine-grained view of how internal citation interacts with retrieved context.

---

### Author Response · Authors · 2025-12-02
**Rebuttal Summary for Area Chair (2/2)**

## 4. Additional clarifications of misunderstandings

> `3iUa-w2` (Scope and stability of CitePretrain and RepliQA)
> Concern that RepliQA might be ingested by future models and that CitePretrainBench does not introduce new raw text.

**Re:**  RepliQA’s authors explicitly request that the dataset **not** be used for pretraining. If an llm provider violates this, it reflects a failure of data curation by that provider, not a flaw of the benchmark. More importantly, the value of CitePretrainBench lies in its **diagnostic power**, not in adding new text. Our benchmark surfaces core failure modes of Passive Index that prior synthetic corpora could not reveal.

---

> `icLP-w2` (Transfer to messy web-scale sources with noisy IDs)
> Concern that the method may not transfer to noisy enterprise or web-scale settings with unstable IDs or missing titles.

**Re:**  Our experiments already include noisy sources like Common Crawl. The method successfully generates natural, deduplicated titles for these sources.

---

> `icLP-w3` (Absolute scores and “citation lags correctness”)
> Concern that citation precision is low in absolute terms and lags behind the model’s own answer correctness.

**Re:**  First, the premise that citation lags behind correctness is incorrect: citation actually matches or exceeds QA accuracy on 3/4 datasets, and the only exception is due to metric mismatch. Second, low absolute numbers reflect **model capacity**, not a limitation of our method. A 7B model that struggles with QA cannot be expected to have near-perfect citation. As we scale from 3B to 7B to 14B, both correctness and citation improve monotonically, and scaling Active Indexing data further boosts a fixed 7B model without saturation. This shows that citation performance is bounded by model ability but that Active Indexing can significantly raise that ceiling.

---

> `icLP-w4` (Hybrid gap to oracle and unresolved conflicts)
> Concern that the hybrid still leaves a noticeable gap to the oracle combination, suggesting conflict resolution between parametric and retrieved knowledge is not fully solved.

**Re:** Fully resolving all knowledge conflicts is outside the scope of this work; our study focuses on internal citation, and this experiment's goal is to demonstrate that internal citation provides a complementary signal to external retrieval.

---

We hope this summary saves the AC time by capturing the core of our rebuttal improvements. Please refer to the individual reviewer responses for granular details.

---

### Meta-Review · Area_Chair_fcs8 · 2026-01-05

**Summary:**

The reviewers consistently identified several strengths of this work including:
1. That the problem and approach are well-motivated (grma, icLP)
2. More than one reviewer referred to the Active Indexing approach as an elegant solution and appreciated the construction of the CitePretrainBench (grma, m27s, icLP)
3. Reviewers also praised the evaluation and ablations for being thorough and providing insights. (grma, m27s, 3iUa)
4. CitePretrainBench is the first to go beyond synthetic examples, to incorporate real-world complexities (m27s)

Reviewers also raised several substantial concerns in their initial reviews:
1. The additional training cost due to augmented data used in Active Indexing (grma, icLP)
2. Concerns about the claims regarding generalization vs memorization, and whether document titles present a shortcut risk, including limited evaluation on OOD examples to truly test generalization (grma, m27s, 3iUa)
3. Limited number of baselines included in the study (m27s, icLP)
4. Not isolating the "conflict slice" (m27s)
5. Novelty and motivation of the proposed approach (m27s, 3iUa)
6. Scope and stability of CitePretrainBench (3iUa)

**Reviewer Concerns:**

From my reading of the reviews, the rebuttal, and the revised paper, all of the concerns have been clearly addressed.
Reviewer grma responded saying as much.
While there is no other response from other reviewers, all of the concerns mentioned above were addressed via edits to the paper and many of these also include additional experiments, (e.g., to evaluate title shortcut risk, analyzing conflict slices, ...).

Hence, from my perspective there are not any outstanding concerns.

**Reviewer Scores:**

Reviewer grma, who provided a detailed and thorough review, responded saying that they would keep their score at 8 Accept.

All of the other reviewers (m27s, 3iUa, and icLP) had initial ratings of 4 Marginally Below, and did not provide any indication/message before the system was closed.

Based on the rebuttal, if I were the one to have written those three reviews, I would have been inclined to increase my score to at least a Marginally Above rating, if not higher.

---

### Decision · Program_Chairs · 2026-01-26

Accept (Poster)